# Geometry Awakening: Cross-Geometry Learning Exhibits Superiority over Individual Structures

**Yadong Sun**[1] **Xiaofeng Cao**[1,✉] **Yu Wang**[1] **Wei Ye**[2] **Jingcai Guo**[3] **Qing Guo**[4]

[1]School of Artificial Intelligence, Jilin University, China
[2]College of Electronic and Information Engineering, Tongji University, China
[3]The Hong Kong Polytechnic University
[4]CFAR and IHPC, Agency for Science, Technology and Research (A*STAR), Singapore
`sunyd22@mails.jlu.edu.cn, xiaofengcao@jlu.edu.cn, yu_w18@mails.jlu.edu.cn,`
`yew@tongji.edu.cn, jc-jingcai.guo@polyu.edu.hk, tsingqguo@ieee.org`

## Abstract

Recent research has underscored the efficacy of Graph Neural Networks (GNNs) in modeling diverse geometric structures within graph data. However, real-world graphs typically exhibit geometrically heterogeneous characteristics, rendering the confinement to a single geometric paradigm insufficient for capturing their intricate structural complexities. To address this limitation, we examine the performance of GNNs across various geometries through the lens of knowledge distillation (KD) and introduce a novel cross-geometric framework. This framework encodes graphs by integrating both Euclidean and hyperbolic geometries in a space-mixing fashion. Our approach employs multiple teacher models, each generating hint embeddings that encapsulate distinct geometric properties. We then implement a structure-wise knowledge transfer module that optimally leverages these embeddings within their respective geometric contexts, thereby enhancing the training efficacy of the student model. Additionally, our framework incorporates a geometric optimization network designed to bridge the distributional disparities among these embeddings. Experimental results demonstrate that our model-agnostic framework more effectively captures topological graph knowledge, resulting in superior performance of the student models when compared to traditional KD methodologies.

## 1 Introduction

Graph Neural Networks (GNNs) have emerged as indispensable tools for analyzing relational data in diverse domains, such as natural language processing [1, 2, 3], computer vision [4, 5], recommendation systems [6, 7]. Their conventional approach of operating within Euclidean space encounters limitations when confronted with datasets embodying non-Euclidean characteristics, such as power-law distribution and hierarchical structures, prevalent in real-world applications [8]. Recognizing this challenge, our community ventures into the realm of non-Euclidean Graph Neural Networks, seeking to harness alternative geometries, notably hyperbolic space, for more adeptly capturing the intricate topological features inherent in many real-world networks [9]. By synthesizing recent advancements and empirical findings, we endeavor to elucidate the potential of non-Euclidean GNNs in effectively modeling complex relational data structures, thereby paving the way for advancements in various application domains [10, 11, 12, 13, 14, 15].

Unlike the constant and flat Euclidean geometry, hyperbolic geometry offers greater flexibility by integrating curvature information, enabling better alignment with the characteristics of non-Euclidean input graphs. This endeavor has rendered hyperbolic GNNs more accessible and comparable to their Euclidean counterparts, resulting in promising performance and interpretability in graph representa-

38th Conference on Neural Information Processing Systems (NeurIPS 2024).

tion learning. Hyperbolic GNNs [16] extended the neighborhood aggregation operation by computing centroids in the hyperbolic geometry. This approach effectively fuses the node features and hierarchical structure, thereby learning superior node representations. Furthermore, a full manifold-preserving feature transformation operation has been developed in hyperbolic geometry [15], eliminating the complicated transformations between hyperbolic and tangent spaces. With these essential operation, hyperbolic GNNs can achieve comparable or even superior performance than Euclidean GNNs.

**Question.** *Although there has been a surge of research on Euclidean and non-Euclidean GNNs in the community, it remains unclear which geometry offers greater advantages. Real-world graphs often exhibit geometrically structural heterogeneity, characterized by variations in clustering and density among nodes [17, 18], as shown in Figure 1. The structural heterogeneity pose a challenge when attempting to accurately model the graph structure using GNNs solely equipped with either Euclidean or non-Euclidean geometry.*

**Motivation.** According *Local Subgraph Preservation Property* [19], the properties of a node largely depend on the properties of the local subgraph centered around it. Considering the hyperbolic property of local subgraphs, i,e., hyperbolicity [1]. Employing hyperbolic geometry modeling achieves higher precision and minimal information loss when hyperbolicity is low. Conversely, when hyperbolicity is elevated, opting for Euclidean geometry modeling results in lower complexity and slightly superior performance compared to hyperbolic geometry. Consequently, the primary limitation of existing graph neural networks is their inability to adaptively select the appropriate geometry for representing nodes with different local structures [21, 22].

**Our scheme.** In this paper, we propose a cross-geometric graph knowledge distillation framework that encodes graphs utilizing both Euclidean and hyperbolic geometry in a locally space-mixing fashion. In contrast to traditional methods that compute hyperbolicity for the overall graph and roughly analyze its applicability to different geometries, our approach performs fine-grained analysis on the local subgraphs surrounding each node. This enables the selection of embeddings in the most appropriate geometry for local subgraphs, which is subsequently utilized to transfer knowledge to the student model. Additionally, we introduce a geometric embedding optimization module to optimize the distribution of embeddings produced by the teacher models. To evaluate the performance of our proposed approach, we conduct distillation experiments on node classification (NC) and link prediction (LP) tasks across various types of graph data. The experimental results demonstrate the superiority of our approach in teaching student models compared to other baseline methods. Our approach highlights enhanced effectiveness and generalization, ultimately achieving state-of-the-art performance in graph data distillation tasks. The salient aspects of our contributions are as follows:

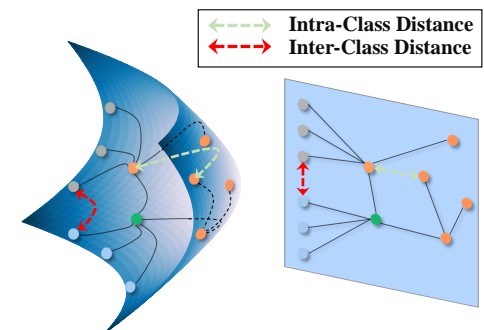

Figure 1: Visualization of the embedding of the same graph in hyperbolic space (left) and Euclidean space (right), with different colors representing different class labels. Tree-like subgraphs maintain significant inter-class margins in hyperbolic space, leading to improved classification boundaries, while intra-class nodes with Euclidean properties may be overstretched due to the utilization of hyperbolic metrics, hence embedding in Euclidean space is preferred.

- Structured analysis reveals that both Euclidean and hyperbolic geometries demonstrate commendable performance in graph processing, despite their inherent disparities and potential geometric conflicts. This prompts scholarly inquiry into reconciling these divergent geometric spaces within GNNs, inspiring new research avenues in geometry awareness.

- With heightened awareness, there's potential to represent graph structures across dimensions, transcending singular space limitations. We thus adapt and integrate teacher embeddings from diverse geometries, transferring them to more effective cross-geometric space.

---

[1]Gromov's $\delta$-hyperbolicity [20] (See the appendix A.1 for the calculating process) measures a graph's tree-like structure, with lower $\delta$ values indicating higher hyperbolicity in a graph dataset, where $\delta = 0$ represents a tree.

- Extensive experiments employing KD techniques on diverse graph datasets demonstrate that cross-geometric methods significantly outperform traditional approaches in the context of knowledge transfer. This is particularly evident in NC and LP tasks, thereby affirming the superior efficacy of these methods.

## 2 Related Work

**Graph Neural Networks.** GNNs are neural network models that capture interdependencies between nodes by propagating messages among them within a graph. The most representative model is Graph Nonvolutional Network (GCN) [23, 24, 25], which can be regarded as a generalization of convolutional neural networks to graph data. The GraphSAGE [26] employs a neighbor sampling strategy to address graph data, enabling information aggregation based on the local neighborhood structure of nodes. The attention-based GNN [27, 28, 29] model employs masked self-attention, assigning diverse weights to node representations based on the varying features of neighboring nodes. Notably, constructing GNNs in the hyperbolic space [13, 15, 30] significantly reduces embedding distortions caused by the inability of Euclidean space to handle power-law distributions, particularly in the case of tree-like or highly hierarchical data.

**Knowledge Distillation.** KD initially proposed by [31], is a model compression technique that involves leveraging pre-trained teacher models to guide the training of a lightweight student model [32, 33]. After that, [34] aligns student and teacher model intermediate features using a regressor and a loss function to minimize feature differences. [35] employs attention mechanisms to extract features from teacher model's intermediate layers and transfer them to the student model. As GNNs have demonstrated breakthrough performance in various deep learning tasks, a number of graph-based KD frameworks have been proposed successively. [36] introduces a local structure preserving module to extract knowledge from intermediate layers of the GNN model, guiding the student model to optimize learned topological structures. [37] proposes a novel approach to effectively learn multi-scale topological semantics from multiple GNN teacher models to guide student model. [38] incorporates a VQ-VAE to learn a codebook that represents informative local structures, and uses these local structures as additional information for distillation. However, all these methods rely on the Euclidean geometry. Our proposed approach leverages both Euclidean and non-Euclidean geometries to learn representations of highly hierarchical local structures, ensuring that the knowledge transmitted to the student model is highly reliable.

## 3 Problem Definition and Preliminaries

### 3.1 Problem Definition

For the graph KD, given a graph $\mathcal{G} = (\mathcal{V}, \mathcal{E})$, where $\mathcal{V}$ denotes the node set and $\mathcal{E}$ denotes the edge set. Let $N$ denotes the total number of nodes in the graph $\mathcal{G}$, $\mathbf{X}$ denotes nodes' feature matrix with each row corresponding to the feature vector of a node, and $\mathbf{A}$ denotes graph's $N \times N$ adjacency matrix, where $A_{ij}$ signifies whether there is an edge between nodes $i$ and $j$. If $A_{ij} = 1$, an edge exists; otherwise, no edge is present. Let $\mathcal{M}_T = \{m_{T_1}, m_{T_2}, ..., m_{T_R}\}$ denotes the teacher models pre-trained on $\mathcal{G}$, $R$ represents the number of geometries. Our fundamental objective is to extract information from $\mathcal{G}$ by $\mathcal{M}_T$, and employing it to boost the training process of the student model, denoted as $m_S$, which in Euclidean space and has significantly smaller size. Let $\mathcal{Z}_T = \{\mathbf{Z}_{T_1}, \mathbf{Z}_{T_2}, ..., \mathbf{Z}_{T_R}, \}$ be the outputs of the teacher models and $\mathbf{Z}_S$ be the outputs of the student model. The optimization goal is to minimize the disparity between predictions of $\mathcal{M}_\mathbf{T}$ and $m_S$ on $\mathcal{G}$, i.e.,

$$\min_{m_s} \frac{1}{N} \sum_{i=1}^{N} \sum_{j=1}^{R} \beta_j \cdot \mathcal{F}_{dis}(\boldsymbol{z}_{T_j,i} || \boldsymbol{z}_{S,i}), \tag{1}$$

where $\beta_j$ denotes the weight of the $j$-th teacher model, $\mathcal{F}_{dis}$ denotes disparity measurement function.

### 3.2 Preliminaries

In this paper, We focus on distillation performance of the teacher model individually in Euclidean, hyperbolic, and spherical geometries, as well as across geometries. We give a necessary introduction of hyperbolic geometry in this subsection, with other information available in Appendix A.

Hyperbolic geometry studies the properties of curved space with negative curvature. Hyperbolic space can be modeled using five isometric models [39, 40], and in this paper, we adopt Poincaré disk model. The Poincaré disk model evinces a distinctive property wherein distances from the geometric center to the periphery undergo a non-linear augmentation as a function of layer depth[9]. This phenomenon engenders a nonlinear and multi-branch composite structure within the model's geometric framework.

**Definition 3.1** (Poincaré Disk Model). *A $n$-dimensional Poincaré disk model $(\mathbb{B}_c^n, g^{\mathbb{B}})$ is a complete Riemannian manifold with a negative constant curvature $c$, which defined as*

$$\mathbb{B}_c^n := \left\{ \boldsymbol{x} \in \mathbb{R}^n : -c\|\boldsymbol{x}\|^2 < 1 \right\}, \quad g^{\mathbb{B}} = (\lambda_{\boldsymbol{x}}^c)^2 g^{\mathbb{E}}, \quad \lambda_{\boldsymbol{x}}^c = \frac{2}{1 - c\|\boldsymbol{x}\|^2} \tag{2}$$

*where $\|\cdot\|$ denotes the Euclidean norm, $g$ denotes the Riemannian metric, and The superscripts $^{\mathbb{B}}$ and $^{\mathbb{E}}$ indicate that the vector or matrix resides in hyperbolic space and Euclidean space, respectively.*

**Definition 3.2** (Hyperbolic Operations). *Given two points $\boldsymbol{x}, \boldsymbol{y} \in \mathbb{B}_c^n$, the hyperbolic distance between them is defined as*

$$d_c(\boldsymbol{x}, \boldsymbol{y}) = \frac{2}{\sqrt{c}} \tanh^{-1}\left(\sqrt{c}\,\|-\boldsymbol{x} \oplus_c \boldsymbol{y}\|\right), \tag{3}$$

*where $\oplus_c$ denotes Möbius addition, i.e.,*

$$\boldsymbol{x} \oplus_c \boldsymbol{y} := \frac{\left(1 + 2c\langle\boldsymbol{x}, \boldsymbol{y}\rangle + c\|\boldsymbol{y}\|^2\right)\boldsymbol{x} + \left(1 - c\|\boldsymbol{x}\|^2\right)\boldsymbol{y}}{1 + 2c\langle\boldsymbol{x}, \boldsymbol{y}\rangle + c^2\|\boldsymbol{x}\|^2\|\boldsymbol{y}\|^2}. \tag{4}$$

**Definition 3.3** (Tangent Space). *The tangent space at a point $\boldsymbol{x}$ in hyperbolic space, denoted as $\mathcal{T}_{\boldsymbol{x}}\mathbb{B}_c^n$, serves as the first-order approximation of the original space, a $n$-dimensional tangent space is isomorphic to Euclidean space $\mathbb{R}^n$. Representations between hyperbolic and tangent space can be transformed via the exponential and logarithmic map as follows:*

$$
\begin{aligned}
\mathcal{T}_{\boldsymbol{x}}\mathbb{B}_c^n \to \mathbb{B}_c^n : \exp_{\boldsymbol{x}}^c(\boldsymbol{v}) &= \boldsymbol{x} \oplus_c \left(\tanh\left(\sqrt{c}\frac{\lambda_{\boldsymbol{x}}^c\|\boldsymbol{v}\|}{2}\right)\frac{\boldsymbol{v}}{\sqrt{c}\|\boldsymbol{v}\|}\right), \\
\mathbb{B}_c^n \to \mathcal{T}_{\boldsymbol{x}}\mathbb{B}_c^n : \log_{\boldsymbol{x}}^c(\boldsymbol{y}) &= d_c(\boldsymbol{x}, \boldsymbol{y})\frac{-\boldsymbol{x} \oplus_c \boldsymbol{y}}{\lambda_{\boldsymbol{x}}^c\|-\boldsymbol{x} \oplus_c \boldsymbol{y}\|},
\end{aligned}
\tag{5}
$$

*where $\boldsymbol{v} \in \mathcal{T}_{\boldsymbol{x}}\mathbb{B}_c^n$, $\boldsymbol{y} \in \mathbb{B}_c^n$ and $\lambda_{\boldsymbol{x}}^c$ has same meaning in Eq. (2). Here we utilize the origin point $\boldsymbol{o}$ in hyperbolic space as a reference point $\boldsymbol{x}$ to equalize errors across various directions.*

## 4 Cross-Geometry Learning with KD

This section reveals three key aspects: why cross-geometry learning demonstrates superior performance, why it is feasible, and how this superiority is achieved by employing KD. Thus, we analyze our method from three perspectives: reasonableness, superiority, and trustworthiness.

### 4.1 Geometric Features of Local Subgraphs

**Reasonableness:** According *local subgraph perservation peoperty theorem* [19], nodes near the central node strongly affect its features, while distant nodes typically have negligible impact. The graph data in real-world often exhibits significant complexity, where diverse local subgraphs often entail distinct geometric properties [17, 18], employing cross-geometry system can offer more effective embedding selections for local graphs, thereby achieving performance beyond that of single geometry methods.

**Definition 4.1** (Subgraphs of Centroid $p$): *For a given node $p$ belonging to graph $\mathcal{G}$, its corresponding $k$-hops subgraph $\mathcal{G}_p$ comprises all nodes $q \in \mathcal{V}\backslash\{p\}$ within a distance no greater than $k$ from $p$, along with their respective edges.*

We employ the $k$-hops neighbors method to generate subgraphs. Hence, we determine the optimal value of $k$ through the statistical analysis of graph data in section 5.4. For each subgraph $\mathcal{G}_i$, we calculate their Gromov's $\delta$-hyperbolicity (See the appendix A.1 for the calculating process) based on $\mathbf{X}$, denoted as $\delta_{\mathcal{G}_i}$, which serves as a geometric characterization metric for the central node $i$.

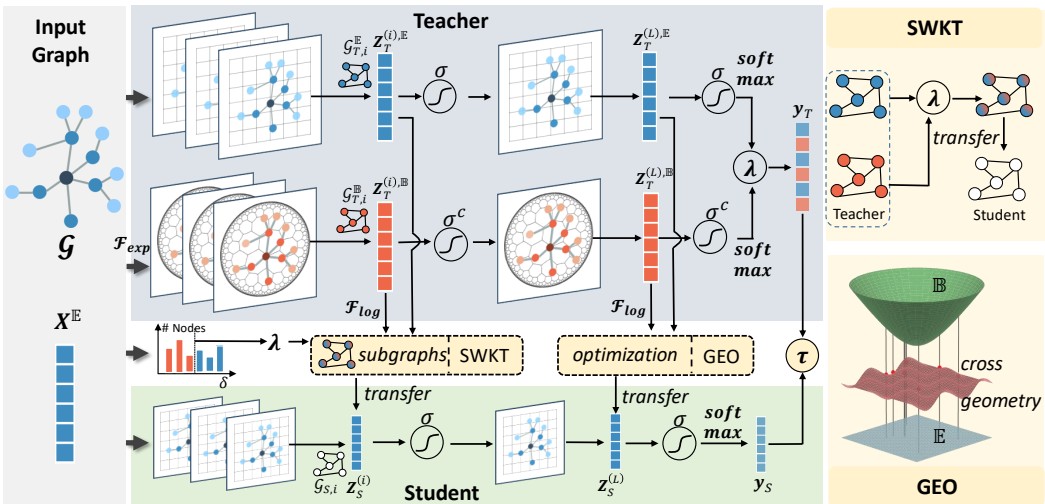

Figure 2: Illustration of our proposed cross-geometry graph KD framework. **Structure-Wise Knowledge Transfer (SWKT)**: Choosing embeddings in appropriate geometric spaces using $\delta_{\mathcal{G}_i}$ of nodes, conveying local subgraph topological knowledge to the student model: $\mathbf{Z}_T^{(i),\mathbb{E}}$ denotes Euclidean teaching, and $\mathbf{Z}_T^{(i),\mathbb{B}}$ denotes the hyperbolic teaching. **GEO**: Enhancing hint embeddings from the teacher models, reducing the negative effects of inconsistencies between different geometries.

## 4.2 Geometric Teacher Models

**Superiority:** We leverage KD technology, utilizing its ability to transfer knowledge between different model architectures, as a medium for interoperability between different geometries. Our proposed KD framework is model-agnostic, making it applicable to various geometric models.

Herein, the framework will be explained using the GCN model [23]. To minimize disparities between the intermediate layers of the teacher and student models, our method uses pre-activation node embeddings $z$ to guide the student model and constructs an embedding matrix $\mathbf{Z}$ for all nodes' $z$. During the forward propagation process of a GCN layer in Euclidean space, the intermediate embeddings of nodes in the $l$-th layer of GCN is given by

$$\mathbf{Z}_T^{(l),\mathbb{E}} = \hat{\mathbf{A}}\mathbf{H}_T^{(l-1),\mathbb{E}}\mathbf{W}^{(l)}, \tag{6}$$

where $\mathbf{H}_T^{(l-1),\mathbb{E}} = activation(\mathbf{Z}_T^{(l-1),\mathbb{E}})$ denotes the node representation matrix from the output of the $(l-1)$-th GCN layer. $\hat{\mathbf{A}}$ denotes the symmetrically normalized adjacency matrix, $\mathbf{W}^{(l)}$ denotes the weight matrix.

For the $i$-th node in layer $l$, its embedding in hyperbolic space is denoted as $z_{T,i}^{(l),\mathbb{B}}$ and its representation is denoted as $h_{T,i}^{(l),\mathbb{B}}$. During the forward propagation process of a GCN layer in hyperbolic space, the transformed feature is given by

$$\boldsymbol{f}_{T,i}^{(l),\mathbb{B}} = \exp_{\boldsymbol{o}}^c\left[\mathbf{W}^{(l)}\log_{\boldsymbol{o}}^c\left(\boldsymbol{h}_{T,i}^{(l-1),\mathbb{B}}\right)\right] \oplus_c \mathbf{b}^{\mathbb{B}}, \tag{7}$$

By performing neighborhood aggregation on these features, we obtain the hyperbolic intermediate embeddings of $i$-th node in the $l$-th layer as follows:

$$\boldsymbol{z}_{T,i}^{(l),\mathbb{B}} = \exp_{\boldsymbol{f}_{T,i}^{(l),\mathbb{B}}}^c\left[\sum\nolimits_{j:(i,j)\in\mathcal{E}} w_{ij}\log_{\boldsymbol{f}_{T,i}^{(l),\mathbb{B}}}^c\left(\boldsymbol{f}_{T,j}^{(l),\mathbb{B}}\right)\right], \tag{8}$$

where $w_{ij}$ is the weight coefficient computed by $\boldsymbol{f}_{T,i}^{(l),\mathbb{B}}$ and $\boldsymbol{f}_{T,j}^{(l),\mathbb{B}}$.

## 4.3 Structure-Wise Knowledge Transfer

**Trustworthiness:** To achieve a more fine-grained selection of embeddings in appropriate geometry, we designed a Structure-Wise Knowledge Transfer (SWKT) module. This module determines suitable

geometric representations based on the geometric feature $\delta_{\mathcal{G}_i}$ of subgraphs and transfers them to the student, facilitating more effective information extraction and guidance.

Specifically, we obtain representations of local subgraphs centered around each node based on the $l$-th intermediate layer hint embeddings of the teacher model in different geometries. We denote $i$-th local structure representation in hyperbolic geometry as $\boldsymbol{u}_{T,i}^{(l),\mathbb{B}} = \{u_{T,i1}^{(l),\mathbb{B}}, u_{T,i2}^{(l),\mathbb{B}}, ..., u_{T,ij}^{(l),\mathbb{B}}, ..., u_{T,iN}^{(l),\mathbb{B}}\}$. Element $u_{T,ij}^{(l),\mathbb{B}}$ in hyperbolic geometry can be computed as follows:

$$
\begin{aligned}
u_{T,ij}^{(l),\mathbb{B}} &= \mathcal{F}_{sub}\left(\log_{\boldsymbol{o}}^c(\boldsymbol{z}_{T,i}^{(l)}), \log_{\boldsymbol{o}}^c(\boldsymbol{z}_{T,j}^{(l)})\right) \\
&= \exp\left(\left\|\log_{\boldsymbol{o}}^c(\boldsymbol{z}_{T,i}^{(l),\mathbb{B}}), \log_{\boldsymbol{o}}^c(\boldsymbol{z}_{T,j}^{(l),\mathbb{B}})\right\|^2\right) / \sum_{j:(j,i)\in\mathcal{E}}\left(\exp\left(\left\|\log_{\boldsymbol{o}}^c(\boldsymbol{z}_{T,i}^{(l),\mathbb{B}}), \log_{\boldsymbol{o}}^c(\boldsymbol{z}_{T,j}^{(l),\mathbb{B}})\right\|^2\right)\right)\cdot
\end{aligned}
\tag{9}
$$

Similarly, we can obtain representation set of $i$-th local structure in Euclidean geometry denoted as $\boldsymbol{u}_{T,i}^{(l),\mathbb{E}}$. SWKT generates induced $k$-hops subgraphs centered at each node, computes their $\delta_{\mathcal{G}_i}$ as the hierarchical level of the central node $i$, and obtains teacher models' middle layer representations of the $i$-th node in the $l$-th layer based on $\delta_{\mathcal{G}_i}$ as follows:

$$
\boldsymbol{u}_{T,i}^{(l)} = \boldsymbol{u}_{T,i}^{(l),\mathbb{B}} \cdot \mathbb{I}(\delta_{\mathcal{G}_i} < \lambda) + \boldsymbol{u}_{T,i}^{(l),\mathbb{E}} \cdot \mathbb{I}(\delta_{\mathcal{G}_i} \geq \lambda)
\tag{10}
$$

where $\mathbb{I}$ denotes indicator function, the threshold $\lambda$ is a hyperparameter that is typically set to a smaller value on graphs with higher $\delta$-hyperbolicity values to achieve better performance.

In our method, embeddings of $l$-th guided middle layers of student model is $\mathbf{Z}_S^{(l),E}$, and applying Eq. (9) likewise yields the student model $i$-th local structure representation $\boldsymbol{u}_{S,i}^{(l)} = \{u_{S,i1}^{(l)}, u_{S,i2}^{(l)}, ..., u_{S,iN}^{(l)}\}$. For node $i$, the similarity between the local structures of the teacher model and the student model is measured as:

$$
\mathcal{P}_i^{(l)} = D_{KL}\left(\boldsymbol{u}_{S,i}^{(l)} \| \boldsymbol{u}_{T,i}^{(l)}\right) = \sum_{j:(j,i)\in\mathcal{E}} u_{S,ij}^{(l)} \log\left(\frac{u_{S,ij}^{(l)}}{u_{T,ij}^{(l)}}\right),
\tag{11}
$$

where $D_{KL}$ represents the Kullback-Leibler divergence.

SWKT minimizes the local structure similarity $\mathcal{P}$ to transfer knowledge from hint embeddings in different geometry to student model. Thus, the loss function for the SWKT module is

$$
\mathcal{L}_{SWKT} = \frac{1}{L}\frac{1}{N}\sum_{l=1}^{L}\sum_{i=1}^{N}\mathcal{P}_i^{(l)},
\tag{12}
$$

where $L$ denotes the total number of intermediate layers used for distillation.

### 4.4 Geometric Embedding Optimization

**Trustworthiness:** Simply concatenating embeddings from Euclidean and hyperbolic teacher models to teach student model can lead to confusion due to geometric inconsistencies. This confusion may result in the student model performing worse than when learning from a single geometric teacher model, as shown in section 5.2. To mitigate the negative effects caused by this inconsistency, we propose a Geometric Embedding Optimization module (GEO) to optimize cross-geometric space.

Specifically, for a given node $i$ from layer $l$, we have its local geometric information $\delta_{\mathcal{G}_i}$ and teacher embeddings from different geometric spaces. We obtain an initial fused feature as follows:

$$
\boldsymbol{e}_{T,i}^{(l)} = \frac{1}{1 + \exp(-(\delta_{\mathcal{G}_i} - \lambda))} \cdot \boldsymbol{z}_{T,i}^{(l),\mathbb{E}} + \frac{\exp(-(\delta_{\mathcal{G}_i} - \lambda))}{1 + \exp(-(\delta_{\mathcal{G}_i} - \lambda))} \cdot \boldsymbol{z}_{T,i}^{(l),\mathbb{B}},
\tag{13}
$$

where $\lambda$ has the same meaning as Eq. (10).

Next, we use a single-layer GCN (which can be replaced by other sufficiently capable networks, such as a Multi-Layer Perceptron (MLP)) to optimize the initially fused features $\boldsymbol{e}_{T,i}^{(l)}$. The optimization network should select loss functions based on the specific downstream task. In this study, we adopt the triplet loss function [41], which enlarges the distance between different-class nodes and reduces the

distance between same-class nodes, to enhance node classification and link prediction performance. To apply triplet loss to graph data, we organize triplets as follows: Given the hint embeddings from teacher, we sample extensive sets of nodes, where each set includes an anchor node, a positive node with the same label as the anchor, and a negative node with a different label.

Assuming $\mathcal{F}_e$ is corresponding function of a pre-trained GEO network with weight matrix $\mathbf{W}_e$, the elements of the local structure vector $\boldsymbol{u}_{T,i}^{(l)}$ for the $i$-th node of layer $l$ can be represented as

$$u_{T,ij}^{(l)} = \mathcal{F}_{sub}(\mathcal{F}_e(\boldsymbol{e}_{T,i}^{(l)}; \mathbf{W}_e), \mathcal{F}_e(\boldsymbol{e}_{T,j}^{(l)}; \mathbf{W}_e)), \tag{14}$$

where $\mathcal{F}_{sub}$ has the same meaning as Eq. (9).

Subsequently, we can reference Eq. (11) to compute the structural similarity between the outputs of GEO and the guided embeddings of $l$-th layer of the student model as:

$$\mathcal{L}_{GEO} = \frac{1}{N} \sum\nolimits_{i=1}^{N} D_{KL}(\boldsymbol{u}_{T,i}^{(l)} \| \boldsymbol{u}_{S,i}^{(l)}). \tag{15}$$

### 4.5 Distillation Framework

In our proposed graph KD approach, the teacher model's early $L-1$ layers guide the student model using the SWKT module, while the $L$-th layer guides via the GEO module. Additionally, the student model also learns the logits distribution of teacher models, i.e., outputs of GEO in last layer. Given the logits $\boldsymbol{y}_T$ from teacher models and the predicted logits $\boldsymbol{y}_S$ from student model, our overall KD loss is as follows:

$$\mathcal{L} = \mathcal{L}_{SWKT} + \mathcal{L}_{CE}(\boldsymbol{y}_T, \boldsymbol{y}_S) + \beta \mathcal{L}_{GEO}, \tag{16}$$

where $\mathcal{L}_{CE}$ denotes the cross-entropy loss function, $\beta$ is a weight coefficient.

The space complexity is $O(ND + |E| + RNH + kN|E|)$, where $N$ is the number of nodes, $D$ is the feature dimension, $|E|$ is the number of edges, $R$ is the number of teacher models, and $k$ is the $k$-pop parameter. For time complexity, forward propagation has a complexity of $O(NH^2 + |E|H)$, local subgraph generation is $O(kN|E|)$, the Structured-Wise Knowledge Transfer module is $O(kNH)$, similarity computation is $O(kN)$, and the Geometric Embedding Optimization module contributes $O(NH^2)$. Thus, the overall time complexity is $O(NH^2 + |E|H + kN(|E| + H))$.

## 5 Experiments

In this section, we first give the experimental setup and baselines. Then we compare our graph KD framework with some baselines on NC and LP tasks. Hyperparameters and ablation analysis also be given. Further experimental results can be found in Appendix C.

### 5.1 Experimental Settings

**Setups**. We preform NC and LP tasks on citation network datasets `Cora` [42], `Citeseer` [43] and `Pubmed` [44], wikipedia-based article hyperlink network dataset `Wiki-CS` [45], and Physics part of the Coauthor dataset `Co-Physics` [46]. The student and teacher models are both GCN composed of two hidden layers and one output layer. The hidden layer node dimensions are 8 for the student and 128 for teachers. The model parameters are uniformly initialized using the Xavier's uniform initialization [47] method The optimizer uses Adam [48] or Riemannian Adam [49]. We set the value of $k$ for $k$-hops subgraphs to 4. To mitigate errors caused by randomness, each F1-score and ROC AUC is the average of 10 experiments with different random seed values.

**Baselines**. To evaluate the performance of our method, we compare it with KD methods formulated in single geometry, including the following methods. **FitNet** [34] utilizes a regressor to align the intermediate features of the student model with those of the teacher model, quantifying the feature discrepancy using L2 distance. **AT** [35] averages attention maps from both teacher and student models' intermediate hidden layers, quantifying differences between them using a designed loss function. **LSP** [36] extracts local structures from both teacher and student models' intermediate feature maps and measures their difference using KL divergence. **MSKD** [37] Utilizes diverse teacher models with varying layers to guide the student model in capturing topology at different scales. **VQG** [38] incorporates a VQ-VAE to learn a codebook that represents informative local structures, and uses

Table 1: F1 scores(%)↑ and ROC AUC(%)↑ of student model distilled by all KD methods for NC and LP tasks. $\mathbb{E}$, $\mathbb{B}$, $\mathbb{S}$ respectively denote the method being in Euclidean, hyperbolic, and spherical spaces, with multiple symbols representing cross geometric space. $\delta$ represents the global hyperbolicity.

| Method | $\mathcal{M}$ | Wiki-CS $\delta=1.0$ | | Co-Physics $\delta=2.5$ | | Pubmed $\delta=3.5$ | | Citeseer $\delta=4.0$ | | Cora $\delta=11.0$ | |
|---|---|---|---|---|---|---|---|---|---|---|---|
| | | NC | LP | NC | LP | NC | LP | NC | LP | NC | LP |
| Teacher | $\mathbb{E}$ | $79.94_{\pm0.16}$ | $93.77_{\pm0.17}$ | $96.75_{\pm0.18}$ | $95.27_{\pm0.08}$ | $82.56_{\pm0.25}$ | $94.91_{\pm0.32}$ | $73.97_{\pm0.09}$ | $95.27_{\pm0.05}$ | $86.98_{\pm0.08}$ | $92.22_{\pm0.27}$ |
| | $\mathbb{B}$ | $81.83_{\pm0.09}$ | $95.11_{\pm0.27}$ | $97.02_{\pm0.19}$ | $98.14_{\pm0.03}$ | $86.24_{\pm0.05}$ | $94.67_{\pm0.10}$ | $81.83_{\pm0.13}$ | $94.34_{\pm0.12}$ | $90.90_{\pm0.18}$ | $91.98_{\pm0.26}$ |
| | $\mathbb{S}$ | $81.61_{\pm0.60}$ | $85.30_{\pm0.08}$ | $96.98_{\pm0.57}$ | $97.74_{\pm0.07}$ | $86.14_{\pm0.38}$ | $94.63_{\pm0.11}$ | $80.37_{\pm0.07}$ | $94.43_{\pm0.29}$ | $89.43_{\pm0.24}$ | $92.52_{\pm0.15}$ |
| FitNet | $\mathbb{E}$ | $67.89_{\pm4.93}$ | $84.51_{\pm0.88}$ | $96.15_{\pm0.16}$ | $90.28_{\pm0.94}$ | $80.71_{\pm4.40}$ | $84.86_{\pm1.66}$ | $68.66_{\pm6.56}$ | $83.39_{\pm1.78}$ | $80.32_{\pm2.99}$ | $81.50_{\pm1.24}$ |
| | $\mathbb{B}$ | $72.59_{\pm1.38}$ | $84.31_{\pm0.68}$ | $96.49_{\pm0.09}$ | $90.10_{\pm0.94}$ | $81.94_{\pm0.09}$ | $85.01_{\pm1.75}$ | $71.11_{\pm1.27}$ | $84.47_{\pm1.34}$ | $83.39_{\pm1.22}$ | $83.00_{\pm3.45}$ |
| | $\mathbb{S}$ | $72.73_{\pm0.02}$ | $62.65_{\pm1.65}$ | $96.67_{\pm0.09}$ | $90.64_{\pm2.29}$ | $81.92_{\pm1.24}$ | $78.43_{\pm2.82}$ | $71.89_{\pm0.08}$ | $77.00_{\pm0.37}$ | $83.28_{\pm0.23}$ | $72.19_{\pm0.35}$ |
| AT | $\mathbb{E}$ | $72.80_{\pm1.39}$ | $84.53_{\pm0.59}$ | $96.48_{\pm0.10}$ | $90.60_{\pm0.60}$ | $81.20_{\pm0.13}$ | $70.24_{\pm2.59}$ | $69.44_{\pm2.74}$ | $82.49_{\pm2.73}$ | $80.49_{\pm1.61}$ | $61.06_{\pm0.31}$ |
| | $\mathbb{B}$ | $71.93_{\pm1.26}$ | $84.76_{\pm0.41}$ | $96.58_{\pm0.01}$ | $89.72_{\pm0.97}$ | $81.31_{\pm2.13}$ | $71.30_{\pm1.95}$ | $71.08_{\pm1.25}$ | $83.25_{\pm2.01}$ | $82.46_{\pm0.82}$ | $83.81_{\pm2.75}$ |
| | $\mathbb{S}$ | $70.71_{\pm0.05}$ | $62.72_{\pm0.26}$ | $96.07_{\pm0.07}$ | $89.55_{\pm1.91}$ | $81.56_{\pm3.25}$ | $78.73_{\pm2.59}$ | $71.95_{\pm0.07}$ | $76.06_{\pm0.31}$ | $83.08_{\pm0.17}$ | $72.72_{\pm0.30}$ |
| LSP | $\mathbb{E}$ | $69.52_{\pm0.79}$ | $84.71_{\pm0.60}$ | $96.52_{\pm0.06}$ | $90.79_{\pm0.55}$ | $81.73_{\pm1.73}$ | $87.26_{\pm0.52}$ | $71.42_{\pm0.98}$ | $83.81_{\pm2.57}$ | $83.34_{\pm1.06}$ | $83.96_{\pm1.41}$ |
| | $\mathbb{B}$ | $69.52_{\pm0.79}$ | $84.21_{\pm0.98}$ | $96.50_{\pm0.08}$ | $90.44_{\pm0.81}$ | $81.72_{\pm0.35}$ | $86.48_{\pm0.58}$ | $71.44_{\pm0.84}$ | $84.17_{\pm1.24}$ | $82.70_{\pm1.00}$ | $82.21_{\pm1.86}$ |
| | $\mathbb{S}$ | $69.13_{\pm0.70}$ | $63.00_{\pm0.26}$ | $96.27_{\pm0.07}$ | $89.05_{\pm2.89}$ | $82.14_{\pm0.23}$ | $78.86_{\pm3.91}$ | $71.88_{\pm0.07}$ | $77.03_{\pm0.30}$ | $82.48_{\pm0.29}$ | $72.89_{\pm0.30}$ |
| MSKD | $\mathbb{E}$ | $70.40_{\pm4.22}$ | $84.81_{\pm0.89}$ | $96.48_{\pm0.09}$ | $90.64_{\pm0.36}$ | $82.04_{\pm0.20}$ | $86.12_{\pm0.63}$ | $71.26_{\pm0.65}$ | $83.77_{\pm1.07}$ | $82.48_{\pm1.29}$ | $84.38_{\pm1.04}$ |
| | $\mathbb{B}$ | $72.56_{\pm1.15}$ | $84.63_{\pm0.45}$ | $96.58_{\pm0.10}$ | $89.70_{\pm0.72}$ | $81.96_{\pm0.37}$ | $86.23_{\pm0.77}$ | $71.47_{\pm1.25}$ | $85.04_{\pm1.44}$ | $82.16_{\pm1.08}$ | $83.86_{\pm1.70}$ |
| | $\mathbb{S}$ | $72.13_{\pm0.03}$ | $62.07_{\pm0.26}$ | $96.27_{\pm0.06}$ | $89.55_{\pm1.95}$ | $82.16_{\pm0.32}$ | $78.60_{\pm3.26}$ | $71.85_{\pm0.09}$ | $76.72_{\pm0.25}$ | $82.86_{\pm0.21}$ | $73.72_{\pm1.94}$ |
| VQG | $\mathbb{E}$ | $72.48_{\pm1.21}$ | $62.71_{\pm0.26}$ | $96.46_{\pm0.09}$ | $89.55_{\pm0.32}$ | $81.49_{\pm0.33}$ | $78.73_{\pm0.26}$ | $70.57_{\pm1.32}$ | $76.07_{\pm0.31}$ | $83.02_{\pm1.80}$ | $72.72_{\pm0.30}$ |
| | $\mathbb{B}$ | $72.91_{\pm2.75}$ | $69.02_{\pm0.29}$ | $96.65_{\pm0.13}$ | $89.05_{\pm0.29}$ | $81.50_{\pm0.32}$ | $80.24_{\pm0.18}$ | $70.92_{\pm0.92}$ | $76.40_{\pm0.71}$ | $83.24_{\pm1.80}$ | $72.30_{\pm2.87}$ |
| | $\mathbb{S}$ | $72.51_{\pm1.35}$ | $65.75_{\pm0.15}$ | $96.64_{\pm0.11}$ | $88.88_{\pm0.33}$ | $81.50_{\pm0.32}$ | $78.10_{\pm0.41}$ | $69.62_{\pm1.91}$ | $74.72_{\pm0.61}$ | $83.15_{\pm0.18}$ | $74.84_{\pm0.27}$ |
| Cross | $\mathbb{E},\mathbb{S}$ | $70.85_{\pm0.51}$ | $61.89_{\pm0.24}$ | $96.07_{\pm0.07}$ | $88.88_{\pm0.33}$ | $80.45_{\pm0.74}$ | $78.89_{\pm2.64}$ | $71.98_{\pm1.21}$ | $76.08_{\pm0.68}$ | $82.89_{\pm1.87}$ | $71.52_{\pm0.57}$ |
| | $\mathbb{B},\mathbb{S}$ | $70.07_{\pm0.67}$ | $62.75_{\pm2.57}$ | $96.17_{\pm0.07}$ | $90.26_{\pm0.26}$ | $82.23_{\pm0.52}$ | $79.74_{\pm0.32}$ | $71.90_{\pm0.05}$ | $74.33_{\pm0.58}$ | $82.74_{\pm2.19}$ | $71.76_{\pm0.42}$ |
| | $\mathbb{E},\mathbb{B},\mathbb{S}$ | $68.70_{\pm0.14}$ | $62.51_{\pm2.59}$ | $96.37_{\pm0.07}$ | $89.35_{\pm0.28}$ | $81.50_{\pm0.32}$ | $77.99_{\pm0.48}$ | $71.77_{\pm1.60}$ | $77.33_{\pm0.30}$ | $83.19_{\pm2.42}$ | $72.89_{\pm0.36}$ |
| Our | $\mathbb{E},\mathbb{B}$ | $\mathbf{74.17_{\pm0.50}}$ | $\mathbf{86.63_{\pm0.31}}$ | $\mathbf{96.87_{\pm0.22}}$ | $\mathbf{91.88_{\pm0.78}}$ | $\mathbf{82.73_{\pm0.23}}$ | $\mathbf{88.32_{\pm0.22}}$ | $\mathbf{72.60_{\pm0.84}}$ | $\mathbf{86.37_{\pm2.14}}$ | $\mathbf{86.05_{\pm0.60}}$ | $\mathbf{86.95_{\pm0.43}}$ |

these local structures as additional information for distillation. To comprehensively demonstrate the superiority of cross-geometry over individual geometry, we conducted experiments for each method separately in Euclidean space $\mathbb{E}$, hyperbolic space $\mathbb{B}$, and spherical space $\mathbb{S}$. Here, we adopt a spherical space $\mathbb{S}$ with curvature of 1, for further details, please refer to the Appendix A.4. We further conducted exploratory experiments on alternative geometric integration approaches based on that proposed in section 4, as illustrated by **Cross** in Table 1.

## 5.2 Node Classification

We use F1 scores as the evaluation metric for node classification task and present results in Table 1. In comparing LSPs across different geometries, we found a counterintuitive outcome. In datasets with low $\delta$-hyperbolicity, hyperbolic LSP was anticipated to outperform Euclidean LSP. However, it performed worse, dropping by 3.05% (`Wiki-CS`). This suggests that even if a teacher model excels in one geometry, its guidance may be less effective when the student model operates in a different geometry. This highlights a significant gap between hint embeddings in different geometries. Despite employing diverse geometries, our method obtains 1.11% average improvement over SOTA baselines and especially 2.66% and 1.37% on `Cora` and `Wiki-CS`, indicating that our method excels on datasets with lower $\delta$-hyperbolicity. Besides, even in graph with high $\delta$-hyperbolicity, where graph exhibit few hierarchical levels, our method consistently achieves higher F1-score compared to student models obtained by baseline KD methods in a single geometrc space.

With the rapid growth of information, real-world graph data is expanding in scale. To demonstrate the effectiveness of our method on large-scale graphs, we evaluated the F1 scores of NC tasks using distilled student models on larger datasets, `Ogbn-Arxiv` (1,166,243 edges, 169,343 nodes) and `Ogbn-Proteins` (39,561,252 edges, 132,534 nodes) [50]. Results in Table 2 show that our method consistently achieves superior distillation performance on these larger datasets.

## 5.3 Link Prediction

We use ROC AUC as the evaluation metric for link prediction task and present results in Table 1. The average ROC AUC showed an improvement of 1.58%, with particularly notable enhancements on datasets `Wiki-CS` and `Cora`, reaching 1.87% and 2.57%, respectively. Employing KD methods solely based on hyperbolic geometry outperformed those exclusively utilizing Euclidean geometry, particularly on the `Citeseer`. Our cross-geometry KD method outperformed SOTA baselines by

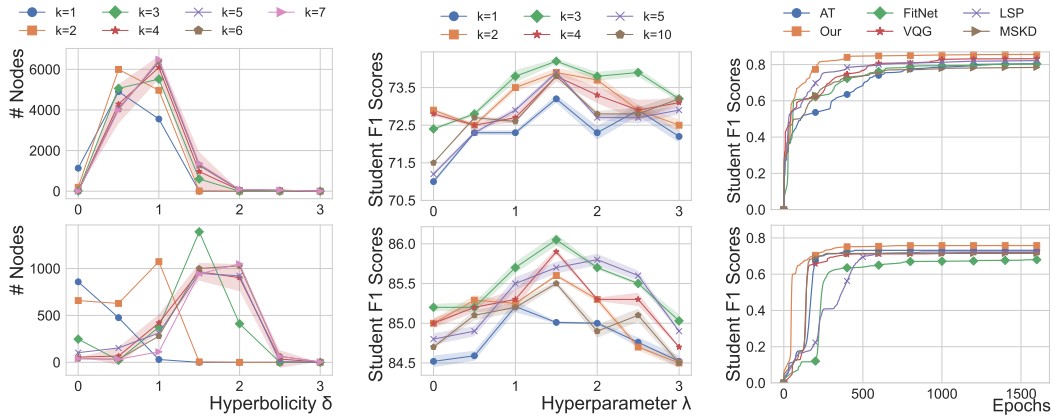

Figure 3: $\delta_{\mathcal{G}_i}$ distribution of subgraphs (left), hyperparameters sensitivity analysis (middle), comparison of convergence rates (right) on the `Cora` (row 1) and `Wiki-CS` (row 2)

0.93% on average, underscoring the efficacy of geometry-specific methods in cross-geometry learning for overall performance enhancement.

## 5.4 Hyperparameters Analysis

$k$ **of Subgraphs**. A small radius $k$ limits local hierarchical assessment, while a large $k$ increases computational complexity. We explored $\delta_i$ value distributions for local geometric properties across $k$ values (1 to 7) on `Wiki-CS` and `Cora` datasets, shown in Figure 3 (left). Stability in distributions occurs at $k \geq 4$, suggesting sufficient capture of geometric characteristics in subgraphs of this size. Thus, we set $k = 4$ for our experiments.

We varied the hyperparameters on the `Wiki-CS` and `Cora` datasets to test the F1 scores for the NC task, with $\lambda \in \{0.0, 0.5, 1.0, 1.5, 2.0, 2.5\}$ , $\beta \in \{1, 2, 3, 4, 5, 10\}$. Here, $\lambda$ denotes the threshold value in Eq. (10) and Eq. (13). A larger $\lambda$ leads to more local subgraphs being embedded in hyperbolic space, while a smaller $\lambda$ results in more subgraphs being embedded in Euclidean space. $\beta$ denotes the weight of the GEO module. The results from Figure 3 (middle) indicate that the hyperparameters $\lambda$ and $\beta$ have a generally minimal impact on the outcomes. The combination of $\lambda = 1.5$ and $\beta = 3.0$ maintains optimal performance.

## 5.5 Distillation Efficiency

To evaluate the convergence efficiency of our proposed KD method, we have recorded the F1-scores trends for student models guided by all KD methods during training epochs on the `Cora` and `Wiki-CS` datasets in Figure 3 (right). As illustrated, our KD method consistently outperform other methods within the same training epochs. Specifically, our KD method achieves state-of-the-art (SOTA) performance before reaching 500 epochs, while the others are still undergoing training. These results serve to validate the effectiveness and efficiency of our method. Due to their simpler architecture, student models generally have faster inference speeds compared to teacher models. The speed-up achieved by student models relative to teacher models is also an indicator of the efficiency of distillation methods. The inference times (in milliseconds) of both teacher and student models, measured on our device, are shown in Table 3. Results demonstrate that our method achieves an average speed-up of approximately 232x.

In addition, we evaluated the training time for each method, the inference time for the corresponding student models, and calculated the compression ratio of student model size relative to the teacher model. The results can be found in Appendix C.

## 5.6 Ablation Study

To further validate the efficacy of cross-geometric learning and the two proposed modules, we conducted additional experiments by adapting our method to operate on a single Euclidean or

Table 2: F1 scores(%)↑ of student model distilled by all KD methods for NC on `Arxiv` and `Proteins`.

| | Teacher / $\mathbb{E}$ | Teacher / $\mathbb{B}$ | FitNet / $\mathbb{E}$ | AT / $\mathbb{E}$ | LSP / $\mathbb{E}$ | MSKD / $\mathbb{E}$ | VQG / $\mathbb{E}$ | Our / $\mathbb{E}, \mathbb{B}$ |
|---|---|---|---|---|---|---|---|---|
| `Arxiv` | 71.91 ±0.06 | 73.21 ±0.19 | 67.56 ±1.79 | 67.48 ±0.25 | 69.53 ±0.03 | 69.27 ±0.21 | 68.59 ±0.11 | **70.89 ±0.53** |
| `Proteins` | 72.83 ±0.09 | 69.23 ±0.02 | 68.71 ±1.81 | 68.53 ±0.35 | 69.45 ±0.23 | 70.97 ±0.97 | 69.54 ±0.36 | **71.22 ±0.41** |

Table 3: Speed-up comparison.

| Datasets | Teacher | Student | Speed-up |
|---|---|---|---|
| `Wiki-CS` | 906 ms | 3.98 ms | 227x |
| `Co-Physics` | 3410 ms | 12.0 ms | 284x |
| `Pubmed` | 914 ms | 4.46 ms | 204x |
| `Citeseer` | 908 ms | 4.01 ms | 226x |
| `Cora` | 975 ms | 4.43 ms | 220x |
| **Average** | 1422 ms | 4.22 ms | 232x |

Table 4: Ablation study results.

| Method | F1 scores (%) | ROC AUC (%) |
|---|---|---|
| w/ Euclidean Teacher | 72.84 ± 1.66 | 84.86 ± 1.02 |
| w/ Hyperbolic Teacher | 72.38 ± 1.83 | 84.55 ± 0.69 |
| w/o SWKT module | 73.40 ± 1.26 | 85.08 ± 0.55 |
| w/o GEO module | 73.39 ± 1.27 | 85.49 ± 0.97 |
| Comprehensive Method | **74.17 ± 0.50** | **86.63 ± 0.31** |

hyperbolic geometry. Additionally, we selectively excluded the SWKT and GEO modules. These ablation experiments were conducted on the `Wiki-CS` dataset, and the results are presented in Table 4. We have the following observations:

- Using only a single geometry, even with the GEO module optimizing embeddings, the enhancement compared to the baseline is minimal.
- The cross-geometric approach consistently outperforms the single-geometric methods. Whether excluding SWKT or GEO, the results are inferior to the comprehensive method, indicating their crucial roles in optimizing geometric embedding distribution.

The overall results of ablation analysis further demonstrate the importance of cross-geometry learning and our proposed two modules.

To demonstrate the model-agnostic nature of our framework, we alter the architecture and the number of layers $L$ in the teacher model. Due to page limitations, we only present key results above. For more details on the dataset, experimental setup, and results, please refer to Appendix B and C.

# 6 Conclusion

Hyperbolic geometry has shown expressive non-Euclidean modeling in the graph community. Noteworthy models, such as Poincaré and Lorentz models, facilitate vector projections between Euclidean and hyperbolic neurons. Our investigation reveals that tree-like or power-law distributed graphs exhibit multiple different hierarchical within locally connected structures. Consequently, training across Euclidean and hyperbolic geometry intuitively emerges as a more flexible approach to graph modeling, yielding significant enhancements in KD tasks. To this end, we introduce a novel KD framework that models the hint embeddings of the teacher models across diverse geometries. By leveraging $\delta$-hyperbolicity, we transfer local subgraphs information to the student model. Our analysis and experiments provide positive support for this innovative perspective on geometry modeling.
**Limitations**. Despite the performance improvements achieved by cross-geometry learning across various tasks, it presents some potential issues. For instance, integrating different geometric information introduces hyperparameters, making the task outcomes somewhat dependent on their selection, thus affecting the method's stability. Additionally, the distillation phase demands more complex pre-trained models, increasing resource and time requirements. These limitations are critical areas for future enhancement in cross-geometry learning.

# 7 Acknowledge

This work was supported by National Natural Science Foundation of China, Grant Number: 62476109, 62206108, 62176184, and the Natural Science Foundation of Jilin Province, Grant Number: 20240101373JC, and Jilin Province Budgetary Capital Construction Fund Plan, Grant Number: 2024C008-5, and Research Project of Jilin Provincial Education Department, Grant Number: JJKH20241285KJ.

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

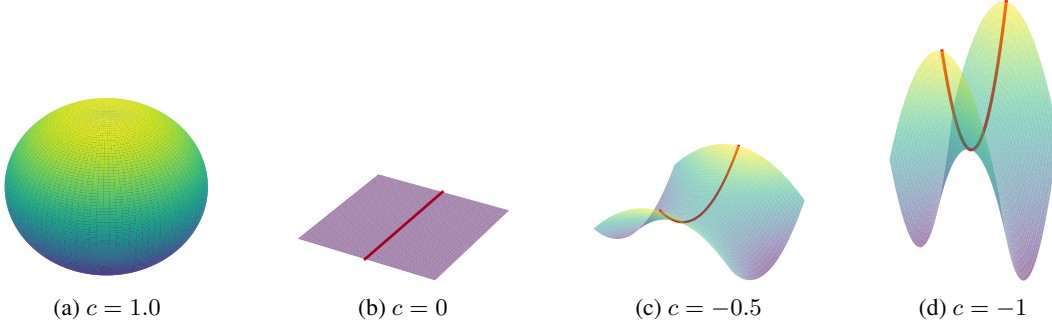

(a) $c = 1.0$      (b) $c = 0$      (c) $c = -0.5$      (d) $c = -1$

Figure 4: Spaces with different curvatures. ($a$) Spherical space with curvature $c = 1.0$. ($b$) Euclidean space. ($c$) Hyperbolic space with curvature $c = -0.5$. ($d$) Hyperbolic space with curvature $c = -1$, which have a faster grow rate of volume.

## A  Additional Theoretical Support

### A.1  Gromov Hyperbolicity

Gromov's $\delta$-hyperbolicity [20] measures a graph's tree-like structure, with lower $\delta$ values indicating higher hyperbolicity in a graph dataset, where $\delta = 0$ represents a tree. In this paper, we compute the hyperbolicity of the $k$-hop subgraph $\mathcal{G}_i$ for each node $i$ as the geometric feature information of the local structure of that node. Here, we provide the detailed calculation process.

First, four nodes $a$, $b$, $c$, $d$ are randomly sampled from the subgraph $\mathcal{G}_i$. Let $S_1$, $S_2$ and $S_3$ be defined as follows:

$$
\begin{aligned}
S_1 &= dist(a,b) + dist(c,d), \\
S_2 &= dist(a,c) + dist(b,d), \\
S_3 &= dist(a,d) + dist(b,c),
\end{aligned}
\tag{17}
$$

where $dist$ denotes shortest path length between two nodes.

Let $M_1$ and $M_2$ be the two largest values among $S_1$, $S_2$ and $S_3$. We define

$$
hyp(a,b,c,d) = M_1 - M_2.
\tag{18}
$$

The hyperbolicity $\delta_{\mathcal{G}_i}$ of the graph $\mathcal{G}_i$ is the maximum of $hyp$ over all possible 4-tuples $(a,b,c,d)$ divided by 2, i.e.,

$$
\delta(G) = \max_{a,b,c,d} \frac{hyp(a,b,c,d)}{2}.
\tag{19}
$$

In our paper, we calculate the $\delta_{\mathcal{G}_i}$ for each k-hop subgraphs. For subgraphs with fewer than four nodes, we label their $\delta_{\mathcal{G}_i}$ value as $N/A$.

### A.2  Euclidean Geometry

Geometry is a branch of mathematics concerned with properties of space such as the distance, shape, size and relative position of figures. In this paper, we analyze the modeling capabilities of graph neural networks in Euclidean , hyperbolic and spherical geometries. Euclidean geometry studies the properties of flat space with zero curvature, where parallel lines never meet, and angles of a triangle sum to 180 degrees. In Euclidean geometry, the volume of space exhibits polynomial growth associated with the dimensionality of the space. The majority of neural network models perform inference operations in this space, where operations such as convolution, pooling, and activation are based on the basic arithmetic operations of addition, subtraction, multiplication, and division.

Euclidean geometry is an axiomatic system, in which all theorems are derived from a small number of simple axioms.

- To draw a straight line from any point to any point.
- To produce (extend) a finite straight line continuously in a straight line.

- To describe a circle with any centre and distance (radius).

- That all right angles are equal to one another.

- [The parallel postulate]: That, if a straight line falling on two straight lines make the interior angles on the same side less than two right angles, the two straight lines, if produced indefinitely, meet on that side on which the angles are less than two right angles.

These axioms provide the fundamental mathematical framework for Euclidean space, allowing GNN models to incorporate information about the absolute positions of nodes, properties of lines, and spatial relationships.

## A.3 Hyperbolic Geometry

Hyperbolic geometry is non-Euclidean geometry, also called Lobachevsky-Bolyai-Gauss geometry. This geometry adheres to all of Euclid's postulates, with the exception of the parallel postulate, which has been substituted with:

- If a straight line intersects two other straight lines, and so makes the two interior angles on one side of it together less than two right angles, then the other straight lines will meet at a point if extended far enough on the side on which the angles are less than two right angles.

Hyperbolic space is a homogeneous space with constant negative curvature. In Euclidean space, the curvature is zero, while in hyperbolic space, the curvature is a negative constant. Moreover, smaller curvature leads to faster volume growth, as illustrated in Figure 4. The hyperbolic space can be modelled using five isomorphic models which are the Lorentz model [51], the Poincaré ball model and Poincaré half space model, and the Klein model. In this paper, we utilize a hyperbolic geometric teacher model based on the Poincaré model. The Poincaré model $\mathbb{B}$ is a manifold equipped with a Riemannian metric $\boldsymbol{g}^B$. This metric is conformal to the Euclidean metric $\boldsymbol{g}^B$. Formally, an $n$ dimensional Poincaré unit ball (manifold) is defined as

$$\mathbb{B}^n = \{x \in \mathbb{R}^n : \|x\| < 1\}, \tag{20}$$

where $\| \cdot \|$ denotes the Euclidean norm. Formally, the distance between $x, y \in \mathbb{B}^n$ is defined as:

$$d(x,y) = arcosh(1 + 2\frac{\|x-y\|^2}{(1-\|x\|^2)(1-\|y\|^2)}). \tag{21}$$

The **Möbius addition** $\oplus$ for $x$ and $y$ in $\mathbb{B}^n$ is defined as

$$x \oplus y = \frac{\left(1 + 2\langle x, y \rangle + \|y\|^2\right) x + \left(1 - \|x\|^2\right) y}{1 + 2\langle x, y \rangle + \|x\|^2 \|y\|^2}. \tag{22}$$

The **Möbius scalar multiplication** $\otimes$ is defined as

$$r \otimes x = \left\{ \begin{array}{ll} \tanh\left(r\operatorname{artanh}(\|x\|)\frac{x}{\|x\|}, & x \in \mathbb{B}^n \\ 0, & x = 0, \end{array} \right., \tag{23}$$

where $r$ is a scalar factor.

The **Möbius vector multiplication** $M^{\otimes}(x)$ is defined as

$$M^{\otimes}(x) = \tanh\left(\frac{\|Mx\|}{\|x\|}\operatorname{actanh}(\|x\|)\right)\frac{Mx}{\|Mx\|} \tag{24}$$

## A.4 Spherical Geometry

Spherical geometry studies the properties of curved space with constant positive curvature, where the angles of a triangle add up to exceeds 180 degrees. All lines in spherical geometry intersect, as there are no parallel lines on a sphere. In the field of graph embedding, spherical geometry plays a significant role as it provides a more realistic model, particularly applicable in geographic information systems and computer graphics. Through spherical geometry, we can accurately describe features on the surface of the Earth and construct data representations with spherical topological structures in

---

**Algorithm 1** Cross-Geometric Graph KD

---

**Input**: Graph $\mathcal{G} = \{\mathcal{V}, \mathcal{E}\}$; Pre-trained teacher $\mathcal{M}_\mathcal{T}$ and GEO model; Initialization parameters $\theta$ of student.
**Parameter**: Threshold $\lambda$; Weight $\beta$.
**Output**: Distilled model's parameter $\theta'$.

 1: **while** student model not converged **do**
 2:    **for** $l$ in $\{1, 2, ..., L\}$ **do**
 3:       Update $\mathbf{Z}^{(l),\mathbb{E}}$ and $\mathbf{Z}^{(l),\mathbb{B}}$, $\mathbf{Z}_T^{(l),\mathbb{E}}$, $\mathbf{Z}_T^{(l),\mathbb{B}} \leftarrow$ Eq. (6), Eq. (8);
 4:       Select appropriate geometry for middle representations, $\boldsymbol{u}_T^{(l)} \leftarrow$ Eq. (10):
 5:       Calculate structure similarity, $\mathcal{P}^{(l)} \leftarrow$ Eq. (11);
 6:       Reduce geometries' discrepancy, Optimize embeddings by Eq. (14);
 7:       $\mathcal{L}_{SWKT}, \mathcal{L}_{GEO} \leftarrow$ Eq. (12), (15);
 8:    **end for**
 9:    Calculate overall Loss by Eq.(16);
10:    Update student model's parameter, $\theta' \leftarrow$ (16);
11: **end while**

---

Table 5: Statistics of datasets.

|  | # Nodes | # Edges | # Features | # Classes | Global Hyperbolicity |
|---|---|---|---|---|---|
| Wiki-CS | 11,701 | 431,726 | 300 | 10 | 1.0 |
| Co-Physics | 34,493 | 495,924 | 8,415 | 5 | 2.5 |
| Pubmed | 19,717 | 88,651 | 500 | 3 | 3.5 |
| Citeseer | 3,327 | 9,928 | 3,703 | 6 | 4.0 |
| Cora | 1,044 | 10,556 | 1,433 | 7 | 11.0 |

three-dimensional space, which is crucial for applications such as map-making, virtual reality, and computer games. In graph embedding, concepts and algorithms of spherical geometry are utilized to process data with spherical topological structures, such as mapping the Earth's surface onto a two-dimensional plane while preserving the correctness of geographic locations and spatial relationships. Therefore, spherical geometry is not only a theoretical discipline but also an indispensable tool in practical applications.

## B  Experiment Details

### B.1  Algorithm

Given the graph data, we initially train teacher models in Euclidean, hyperbolic, and spherical spaces, respectively. Subsequently, we train the MLP model of the GEO module, initializing the student model randomly. By inputting learning parameters alongside hyperparameters $\lambda$ and $\beta$, we employ Algorithm 1 to obtain the distilled student model.

### B.2  Datasets

Here, we present detailed information for each dataset in Table 5. Wiki-CS consists of 11,701 nodes with 431,726 edges, each node characterized by a 300-dimensional feature, and the node labels are categorized into 10 classes. Cora consists of 1,044 nodes with 10,556 edges, each node characterized by a 1,433-dimensional feature, and the node labels are categorized into 7 classes. Pubmed consists of 19,717 nodes with 88,651 edges, each node characterized by a 500-dimensional feature, and the node labels are categorized into 3 classes. Co-Physics consists of 34,493 nodes with 495,924 edges, each node characterized by a 8,415-dimensional feature, and the node labels are categorized into 5 classes. Citeseer consists of 3,327 nodes with 9,928 edges, each node characterized by a 3,703-dimensional feature, and the node labels are categorized into 6 classes. To ensure fairness, we uniformly apply standard splits (70%/15%/15%) for node classification tasks and standard splits (85%/5%/10%) for link prediction tasks.

Table 6: Parameter Settings in NC task.

| Parameters | Wiki-CS | Co-Physics | Pubmed | Citeseer | Cora |
|---|---|---|---|---|---|
| # layers | 2 | 2 | 2 | 2 | 2 |
| teacher hidden dim | 128 | 128 | 128 | 128 | 128 |
| student hidden dim | 8 | 8 | 8 | 16 | 8 |
| learning rate | 0.01 | 0.01 | 0.01 | 0.05 | 0.01 |
| weight decay | 0.0000 | 0.0000 | 0.0005 | 0.0001 | 0.0000 |
| dropout | 0.00 | 0.00 | 0.00 | 0.00 | 0.00 |
| $\lambda$ | 1.5 | 1.5 | 1.5 | 1.5 | 1.5 |
| $\beta$ | 3.0 | 3.0 | 3.0 | 3.0 | 3.0 |

Table 7: Parameter Settings in LP task.

| Parameters | Wiki-CS | Co-Physics | Pubmed | Citeseer | Cora |
|---|---|---|---|---|---|
| # layers | 2 | 2 | 2 | 2 | 2 |
| teacher hidden dim | 128 | 128 | 128 | 128 | 128 |
| student hidden dim | 8 | 8 | 8 | 8 | 8 |
| learning rate | 0.01 | 0.01 | 0.01 | 0.01 | 0.01 |
| weight decay | 0.0000 | 0.0000 | 0.0000 | 0.0000 | 0.0000 |
| dropout | 0.00 | 0.00 | 0.00 | 0.00 | 0.00 |
| $r$ (in fd decoder) | 2.00 | 2.00 | 2.00 | 2.00 | 2.00 |
| t (in fd decoder) | 1.00 | 1.00 | 1.00 | 1.00 | 1.00 |
| $\lambda$ | 1.5 | 1.5 | 1.5 | 1.5 | 1.5 |
| $\beta$ | 3.0 | 3.0 | 3.0 | 3.0 | 3.0 |

## B.3   Setups

For a fair comparison, all methods employ identical teacher and student model architectures on the same dataset. All methods use GCN as the Euclidean teacher model and HGCN as the hyperbolic teacher model. The teacher models consist of two hidden layers and one output layer, with a hidden feature dimension of 128. The student GCN model has two hidden layers and one output layer, with a hidden dimension of 8. During training, the optimizer uses Adam or Riemannian Adam, and hyperparameters such as learning rate, weight decay, and hierarchy threshold are fine-tuned based on the performance of student models on validation sets of different datasets, maintaining consistent hyperparameters for different methods on the same dataset. The parameter configurations for NC are detailed in Table 6, while those for LP are delineated in Table 7. The model parameters are uniformly initialized using the Xavier's uniform initialization method, with a random seed chosen from the range of 0 to 1000. The geo model is trained for 300 epochs, and random sampling during its optimization process involves extracting 100 sets of node pairs for each class.

**Environments**. The running environment includes an Intel Core Intel i7-13700KF CPU with a clock speed of 3.40GHz, boasting 16 cores and 24 threads. A robust NVIDIA GeForce RTX 4070Ti GPU, featuring 12GB of VRAM, encompasses 7680 CUDA cores. The system is equipped with 16GB of RAM. The operating system is Windows 11, and Python 3.10 serves as the programming language. For deep learning tasks, PyTorch version 1.13 is employed, while CUDA version 12.2 enhances GPU acceleration. Package management is facilitated through the use of Anaconda. For large datasets, `Pubmed` and `CoauthorPhysics`, experiments were conducted on a high-performance server with the following specifications: 4 Intel Xeon Gold 5220 CPUs running at 2.20GHz, equipped with 72 cores and 144 threads. The system features 4 Quadro RTX 6000 GPUs, each boasting 24GB of VRAM and 4608 CUDA cores. The system boasts 500GB of RAM and runs on Ubuntu 18.04.6.

Table 8: F1 cores (%)↑ of student models distilled from GAT teacher models on the NC Task.

| Method | $\mathcal{M}$ | Wiki-CS $\delta = 1.0$ | Co-Physics $\delta = 2.5$ | Pubmed $\delta = 3.5$ | Citeseer $\delta = 4.0$ | Cora $\delta = 11.0$ |
|---|---|---|---|---|---|---|
| Teacher | $\mathbb{E}$ | 80.52± 0.23 | 96.86± 0.17 | 84.78± 0.11 | 75.24± 0.18 | 90.07± 0.09 |
| | $\mathbb{B}$ | 82.46± 0.27 | 97.13± 0.23 | 87.35± 0.16 | 82.99± 0.15 | 90.66± 0.05 |
| | $\mathbb{S}$ | 81.96± 0.18 | 96.79± 0.14 | 87.31± 0.08 | 81.97± 0.23 | 89.97± 0.13 |
| Cross | $\mathbb{E},\mathbb{S}$ | 71.75 ± 0.12 | 96.13 ± 0.01 | 80.89 ± 0.55 | 72.11 ± 0.21 | 83.35 ± 0.16 |
| | $\mathbb{B},\mathbb{S}$ | 71.27 ± 0.51 | 96.27 ± 0.07 | 82.25 ± 0.36 | 72.13 ± 0.15 | 83.74 ± 0.37 |
| | $\mathbb{E},\mathbb{B},\mathbb{S}$ | 69.12 ± 0.37 | 96.21 ± 0.02 | 82.51 ± 0.42 | 72.25 ± 0.28 | 83.27 ± 0.25 |
| Our | $\mathbb{E},\mathbb{B}$ | **74.52 ± 0.79** | **97.01 ± 0.04** | **83.46 ± 0.56** | **72.89 ± 0.08** | **86.75 ± 0.48** |

Table 9: F1 Scores (%)↑ of student models distilled from GTN teacher models on the NC Task.

| | Teacher | | GTN Student | | GCN Student | |
|---|---|---|---|---|---|---|
| | $\mathbb{E}$ | $\mathbb{B}$ | F1 Scores | Inference Time | F1 Scores | Inference Time |
| Wiki-CS | 82.74 | 81.83 | 82.48 | 15.34 ms | 74.26 | 3.98 ms |
| Cora | 87.87 | 90.90 | 86.37 | 17.67 ms | 86.24 | 4.43 ms |

## C More Experiment Results and Analysis

### C.1 Replacing Teacher Models

Our proposed framework is model-agnostic. To validate its universality and effectiveness, we conducted experiments by replacing the teacher model from GCN to GAT. These experiments were conducted across three geometries: Euclidean, hyperbolic, and spherical, for cross-geometry learning. The experimental results are presented in Table 8. As depicted in the table, even when the teacher model is replaced with other models, our framework consistently maintains a strong distillation effect, with the combination of hyperbolic and Euclidean geometries still proving to be optimal. Moreover, as the performance of the teacher model improves, there is a corresponding enhancement in the performance of the student model.

We also replaced the Euclidean teacher model with the Graph Transformer Network [52]. The results are shown in Table 9. GTN teacher has 4 layers and a hidden dimension of 128. Specifically, during distillation, student model's $l$ layers match the last $l$ layers of teacher accordingly. The GTN and HGCN teacher output intermediate representations from each layer to the SWKT module for local subgraph structure extracting and selection. These distributions are then optimized by GEO module. Then, these features extracted from the optimized cross-geometric intermediate representations are transferred to students via the corresponding loss function. Additionally, traditional KD loss is computed from the logits output by both the teacher and student models.

### C.2 Replacing Student Models

The student model can operate in other geometric spaces. At first, we chose a Euclidean student model to combine hyperbolic accuracy benefits with Euclidean efficiency and stability. Our framework is model-agnostic, allowing replacement of the student model with other neural networks.To validate student on various geometric spaces, we tested NC F1 scores (%) on the Cora dataset in Table 10. Euclidean and hyperbolic teachers' F1 score is 86.98% and 90.90%.

we conducted experiments using student models with the same architecture as the teacher models in our method. Following results are NC F1 scores (%) on the Cora dataset. Euclidean and hyperbolic teachers' F1 score is 86.98% and 90.90%. The results are shown in Table 11 Compared to the results presented in Table 1 of our paper, student models now even outperform some teacher models, but using the same architecture as the teacher makes student models larger and slower, limiting their suitability for resource-constrained devices.

Table 10: F1 cores (%)↑ of student models in different geometry on the NC Task.

|  | FitNet | AT | LSP | MSKD | VQG | **Our** |
|---|---|---|---|---|---|---|
| Euclidean Studnet | 80.32 | 80.49 | 83.34 | 82.48 | 83.02 | **86.05** |
| Hyperbolic Student | 86.73 | 86.00 | 87.96 | 88.21 | 88.45 | **90.42** |
| Spherical Student | 75.92 | 83.54 | 84.77 | 85.26 | 83.54 | **86.73** |
| Average | 80.99 | 83.34 | 85.35 | 85.31 | 85.00 | **87.73** |

Table 11: F1 Scores (%)↑ of student models with the same architecture as the teacher models on the NC Task.

|  | FitNet | AT | LSP | MSKD | VQG | Our |
|---|---|---|---|---|---|---|
| Student in paper | 80.32 | 80.49 | 83.34 | 82.48 | 83.02 | 86.02 |
| Euclidean student | 86.73 | 86.24 | 86.98 | 86.98 | 86.49 | 87.47 |
| Hyperbolic student | N/A | N/A | N/A | N/A | N/A | 90.42 |

## C.3 Changing Teacher Layers

In addition to model-agnostic features, our framework demonstrates excellent scalability. All experiments in this study were conducted with both teacher and student models having a hidden layer depth of 2. To verify scalability, we configured four types of teacher models, varying their hidden layer depths from 1 to 4, while keeping all other settings constant. By applying the SWKT and GEO modules to each layer, we expanded our framework, as illustrated in Table 12. As observed in the table, despite variations in the number of layers in the teacher models, our framework consistently achieves effective distillation results, with the combination of hyperbolic and Euclidean components remaining optimal. Furthermore, as the performance of the teacher models improves, there is a corresponding enhancement in the performance of the student models.

## C.4 Embeddings Optimization Results

We reduced the embeddings of student models to 2-dimensional space by t-SNE and visualized them in Figure 5, our method yields a superior embedding distribution, which more suitable for NC task.

## C.5 Hyperparameters Analysis

We conducted a more comprehensive hyperparameter analysis on the `Co-Physics`, `Pubmed`, and `CiteSeer` datasets. By adjusting the hyperparameters, we evaluated the F1 scores for the NC task, with $\lambda \in 0.0, 0.5, 1.0, 1.5, 2.0, 2.5$ and $\beta \in 1, 2, 3, 4, 5, 10$. The results, as shown in Figure 6, indicate that the hyperparameters $\lambda$ and $\beta$ have a minimal overall impact on the outcomes. The performance is generally optimal when $\beta = 3$, and $\lambda$ shows better performance at intermediate values.

## C.6 Ablation Study

Following the ablation experiment strategy outlined in Section 5.2, we conducted NC experiments on five other datasets. The results, presented in Table 13, reveal that the performance of the comprehensive method consistently outperforms other conditions across all datasets.

Table 14: Training time spent per epoch (in $ms$) for NC task.

| Datasets | AT | FitNet | LSP | MSKD | VQG | Our |
|---|---|---|---|---|---|---|
| Wiki-CS | 5.25 | 5.50 | 5.18 | 5.27 | 5.26 | 5.16 |
| Cora | 5.22 | 6.38 | 6.09 | 6.03 | 6.01 | 5.81 |
| Pubmed | 5.43 | 6.16 | 5.94 | 6.01 | 5.21 | 6.31 |
| Citeseer | 5.39 | 6.89 | 5.78 | 5.78 | 5.56 | 5.95 |
| Co-Physics | 10.5 | 11.2 | 10.5 | 10.5 | 10.5 | 10.3 |

Table 12: F1 cores (%)↑ of student models distilled from teacher models with different layers on the NC Task.

| Method | $\mathcal{M}$ | $L \times 1$ | $L \times 2$ | $L \times 3$ | $L \times 4$ |
|---|---|---|---|---|---|
| Teacher | $\mathbb{E}$ | $68.42 \pm 0.12$ | $79.94 \pm 0.16$ | $81.22 \pm 0.38$ | $78.12 \pm 0.35$ |
| | $\mathbb{B}$ | $70.04 \pm 0.35$ | $81.83 \pm 0.09$ | $82.73 \pm 0.42$ | $80.45 \pm 0.18$ |
| | $\mathbb{S}$ | $70.15 \pm 0.28$ | $81.61 \pm 0.60$ | $82.51 \pm 0.37$ | $79.86 \pm 0.33$ |
| Cross | $\mathbb{E},\mathbb{S}$ | $51.42 \pm 0.79$ | $70.85 \pm 0.51$ | $71.32 \pm 0.76$ | $69.21 \pm 1.18$ |
| | $\mathbb{B},\mathbb{S}$ | $52.43 \pm 0.52$ | $70.07 \pm 0.67$ | $70.72 \pm 1.35$ | $69.89 \pm 1.21$ |
| | $\mathbb{E},\mathbb{B},\mathbb{S}$ | $52.72 \pm 1.15$ | $68.70 \pm 0.14$ | $69.28 \pm 1.17$ | $68.35 \pm 0.48$ |
| Our | $\mathbb{E},\mathbb{B}$ | $\mathbf{58.12 \pm 2.38}$ | $\mathbf{74.17 \pm 0.50}$ | $\mathbf{74.22 \pm 1.24}$ | $\mathbf{73.75 \pm 1.18}$ |

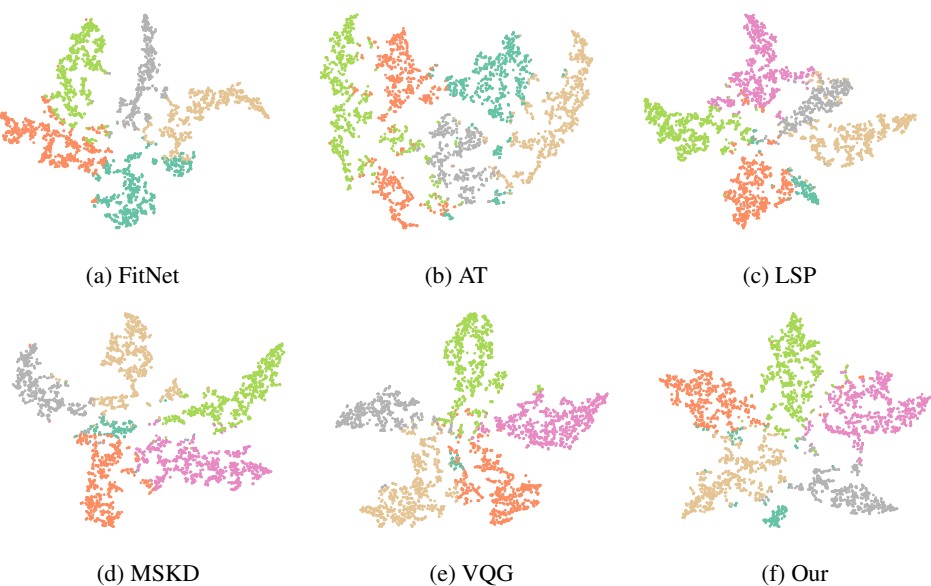

|              |             |            |
|:------------:|:-----------:|:----------:|
| (a) FitNet   | (b) AT      | (c) LSP    |
| (d) MSKD     | (e) VQG     | (f) Our    |

Figure 5: t-SNE Visualization of embeddings obtained by student models.In contrast to baselines,our method achieves embeddings that fully utilize the entire space, ensuring substantial inter-class distances and thereby enhancing node classification performance.

## C.7 Distillation Efficiency

In dataset `Wiki-CS`, we assessed the training and inference time per epoch and the ratio of the total parameter count of the student model to that of the teacher model, as outlined in Table 15. Our KD method achieves similar time efficiency in both training and inference stages compared to other methods. Notably, our method achieves superior results at the highest compression level, thereby further validating the efficacy of our KD method in generating compact yet high-performing student models.

We provide an analysis of the time spent by each knowledge distillation (KD) method during the training of student models on the network classification (NC) task, recorded for every epoch across all datasets, as shown in Table 16. We also present the time taken for inference of the student models on the NC task in Table 14. From the results, it is clear that in various scenarios, the time required for our method is comparable to that of other methods, with no significant increase in time cost. This suggests that our approach effectively balances performance and computational efficiency, making it a practical choice for applications in this field.

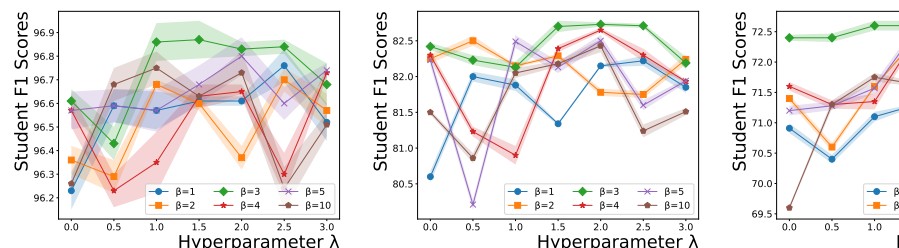

Figure 6: Hyperparameters sensitivity analysis on `Co-Physics` (left), `Pubmed` (middle) and `Citeseer` (right)

Table 13: Ablation experiments for NC task across all datasets, evaluated using F1 score(%)↑.

| Method | Wiki-CS | Cora | Pubmed | Co-Physics | Citeseer |
|---|---|---|---|---|---|
| w/ Euclidean Teacher | $72.84 \pm 1.66$ | $84.55 \pm 0.73$ | $81.85 \pm 0.26$ | $96.50 \pm 0.15$ | $71.16 \pm 1.13$ |
| w/ Hyperbolic Teacher | $72.38 \pm 1.83$ | $84.43 \pm 0.82$ | $81.55 \pm 1.71$ | $96.56 \pm 0.11$ | $71.30 \pm 1.64$ |
| W/o SWKT module | $73.40 \pm 1.26$ | $84.16 \pm 0.89$ | $82.12 \pm 0.41$ | $96.68 \pm 0.13$ | $70.72 \pm 2.62$ |
| w/o GEO module | $73.39 \pm 1.27$ | $84.33 \pm 0.73$ | $82.26 \pm 0.35$ | $96.65 \pm 0.13$ | $71.24 \pm 1.46$ |
| Comprehensive Method | $\mathbf{74.17 \pm 0.50}$ | $\mathbf{84.84 \pm 0.60}$ | $\mathbf{82.61 \pm 0.23}$ | $\mathbf{96.87 \pm 0.22}$ | $\mathbf{72.60 \pm 0.84}$ |

## C.8 Failed Teachers

The failure of one or more teacher models could potentially impact the student model's performance, we have implemented several mechanisms in our method to mitigate this risk:

- **Ensemble Learning**: Using multiple teacher models that capture different geometric properties provides redundancy and robustness. If one model fails, the others still contribute valuable insights, minimizing the impact on the student model.

- **Geometric Optimization Network**: GEO dynamically adjusts the weight of information from each teacher model based on the loss function, reducing the influence of any underperforming model and ensuring the student model receives the most reliable information.

We designed various experimental strategies to assess the impact of failing teachers on students:

- S1: Train student models without KD.
- S2: Train student models with the best-tuned teacher model.
- S3: Train student models with an underperforming teacher model.
- S4: Train student models with an untrained teacher model.
- S5: Train student models with all untrained teacher models.

Note: All methods except MSKD and ours use a single teacher model; thus, S5 is N/A.

Table 15: Time spent per epoch and compression ratio.

| Method | Training (ms) | | Inference (ms) | | Ratio (%) |
|---|---|---|---|---|---|
| | NC | LP | NC | LP | |
| FitNet | 5.50 | 305.7 | 3.98 | 22.92 | 4.47 |
| AT | 5.25 | 314.9 | 3.98 | 22.62 | 4.56 |
| LSP | 5.18 | 318.8 | 3.98 | 22.92 | 4.56 |
| MSKD | 5.27 | 311.4 | 3.98 | 23.94 | 2.67 |
| VQG | 5.23 | 312.5 | 3.98 | 22.60 | 4.47 |
| Our | 5.16 | 305.1 | 3.98 | 23.03 | 2.28 |

Table 16: Inference time spent (in $ms$) for NC task.

| Datasets | AT | FitNet | LSP | MSKD | VQG | Our |
|---|---|---|---|---|---|---|
| Wiki-CS | 3.98 | 3.98 | 3.98 | 3.98 | 3.98 | 3.98 |
| Cora | 4.43 | 4.43 | 4.43 | 4.43 | 4.43 | 4.43 |
| Pubmed | 4.46 | 4.46 | 4.46 | 4.46 | 4.46 | 4.46 |
| Citeseer | 4.01 | 4.01 | 4.01 | 4.01 | 4.01 | 4.01 |
| Co-Physics | 12.0 | 12.0 | 12.0 | 12.0 | 12.0 | 12.0 |

Table 17: F1 scores(%)↑ of student model distilled by all KD methods for NC under failed teachers.

| Methods | S1 | S2 | S3 | S4 | S5 | Std(S2-5) |
|---|---|---|---|---|---|---|
| Teacher($\mathbb{E}$) | N/A | 86.98 | 66.09 | 36.61 | 25.55 | 24.21 |
| Teacher($\mathbb{B}$) | N/A | 90.90 | 64.86 | 52.58 | 30.13 | 21.94 |
| FitNet | 82.06 | 80.32 | 56.27 | 53.81 | N/A | 11.95 |
| AT | 82.06 | 80.49 | 63.39 | 51.11 | N/A | 12.04 |
| LSP | 82.06 | 83.34 | 71.33 | 56.57 | N/A | 10.86 |
| MSKD | 82.06 | 82.48 | 81.82 | 78.62 | 21.62 | 1.68 |
| VQG | 82.06 | 83.02 | 70.02 | 56.02 | N/A | 11.02 |
| Our | 82.06 | 86.05 | 85.26 | 81.33 | 22.85 | 2.06 |

NC f1 score (%) of student models on Cora under different experimental strategies are shown in Table 17. Except S5, which teachers have an average performance of only 30%, our method's distilled student models consistently maintain stable performance even when some teacher models fail.

