# OpenReview forum: "Geometry Awakening: Cross-Geometry Learning Exhibits Superiority over Individual Structures"
_NeurIPS.cc/2024/Conference — NeurIPS 2024 poster_

### Official Review · Reviewer_rr98 · 2024-07-10

**Soundness:** 3
**Presentation:** 3
**Contribution:** 3
**Rating:** 6
**Confidence:** 3

**Summary:**

This paper proposes a novel method for graph knowledge distillation. The proposed method incorporates knowledge from both Euclidean and hyperbolic teacher models and transfers it to a student model in a way that leverages the appropriate geometry for different local subgraphs. A SWKT module is used  to select embeddings from the most suitable geometry based on the local subgraph structure. Furthermore, a GEO module refines the transferred knowledge by mitigating inconsistencies between Euclidean and hyperbolic spaces.
Experimental results demonstrate that the proposed method outperforms other KD methods.

**Strengths:**

S1. The authors propose a method that leverages the strengths of both Euclidean and hyperbolic representations.

S2. The experimental results demonstrate that the proposed method outperforms various graph data distillation baselines.

S3. The proposed method has a similar running time as other baselines with a much higher compression performance.

**Weaknesses:**

W1. The paper would benefit from a thorough analysis of the method's time and space complexity.

W2. The proposed approach uses two geometries (,i.e., Euclidean and hyperbolic). While this is a significant advancement, the paper could be strengthened by discussing its potential extension to incorporate additional non-Euclidean geometries. Exploring the necessary modifications for handling multiple geometries would broaden the applicability of the method.

W3. While providing background knowledge can be helpful, the paper should ensure all definitions in the preliminaries section are essential for understanding the core method. Definitions like the Poincaré Disk Model and Tangent Space, if not directly used, could be moved to an appendix or omitted for clarity.

W4. The paper primarily focuses on Euclidean and hyperbolic teacher models. Further exploration with teacher models employing different geometries, such as spherical space, would provide valuable insights into the method's adaptability. Analyzing the performance with these variations would strengthen the paper's contribution.

**Questions:**

See weaknesses.

**Limitations:**

The authors have discussed the limitations.

---

> ### Author Rebuttal · Authors · 2024-08-06
>
> We thank the reviewer for the positive comments! We hope that our response can resolve your concerns. Please feel free to ask any follow-up questions.
>
> ---
> # W1:
>
> **Notation Definitions:**
> - $N$: Number of nodes
> - $|E|$: Number of edges
> - $D$: Dimension of node features
> - $H$: Dimension of hidden layers
> - $R$: Number of teachers
> - $k$: Parameter for $k$-pop subgraphs
>
> **Space Complexity:**
> - Node feature matrix: $O(N \cdot D)$
> - Adjacency matrix: $O(|E|)$ (assuming the graph is sparse)
> - Representations of hidden layers:  $O(R \cdot N \cdot H)$
> - Optimized distribution of local subgraph representations: $O(N \cdot k \cdot |E|)$
>
> **Overall space complexity: $O(N \cdot D + |E| + R \cdot N \cdot H + N \cdot k \cdot |E|)$**
>
> **Time Complexity:**
> - Forward propagation: $O(N \cdot H^2 + |E| \cdot H)$
> - Local subgraph generation: $O(N \cdot k \cdot |E|)$
> - Structured-WiseKnowledge Transfer (SWKT) module: $O(N \cdot k \cdot H)$
> - Similarity measurement computation: $O(N \cdot k)$
> - Geometric Embedding Optimization (GEO) module: $O(N \cdot H^2)$
>
> **Overall time complexity: $O(N \cdot H^2 + |E| \cdot H + N \cdot k \cdot |E| + N \cdot k \cdot H + N \cdot k)$**
>
> Besides, we also conducted a computational efficiency study comparing the teacher and student models from both theoretical and empirical perspectives in our response to **W1 commented by Reviewer GamE**. In Table 2 of our paper, we present the parameter ratios (compression rates) between the teacher and student models for each knowledge distillation (KD) method. Additionally, we provide information on distillation training times and student inference times in Tables 2, 10, and 11.
>
> ---
> # W2:
>
> We fully agree with your suggestion that exploring the necessary modifications for handling multiple geometries would broaden the applicability of our method. In our paper, **we have simply attempted experiments incorporating spherical geometry**.
>
> In Table 1, the Cross method represent simplified variants of our framework. The rows marked $\mathbb{E}$ and $\mathbb{S}$ correspond to the use of both Euclidean and spherical geometries, while rows $\mathbb{E}$, $\mathbb{B}$, and $\mathbb{S}$ represent the use of Euclidean, hyperbolic, and spherical geometries simultaneously. The results indicate that spherical geometry does not provide significant additional benefits within our framework. We apologize for any confusion caused by unclear descriptions about Table 1 of our paper. We will improve the presentation of the experimental section to enhance clarity.
>
> In future work, we will explore feasible approaches to extend our framework to other potential geometries. For example, we may consider integrating more complex Riemannian geometries or other non-Euclidean geometries to further enhance the performance and applicability of the method. We appreciate your valuable suggestion and look forward to exploring these aspects in our future research.
>
> ---
> # W3:
>
> Yes, moving preliminary definitions that are not directly related to the core method to the appendix will enhance the clarity of the paper.
>
> After carefully review, we found that the expression for the Poincaré Disk Model in Definition 1 is not directly used in the presentation of our method, so we will move it to the appendix. The description of our method requires the projection formula between the tangent space and hyperbolic space in Definition 3, and this conversion involves hyperbolic operations described in Definition 2. Therefore, we will retain Definitions 2 and 3 in the main paper.
>
> Thank you for your constructive suggestion. We will make the necessary adjustments based on your feedback to improve the clarity and readability of the paper.
>
> ---
> # W4:
>
> In fact, **we have conducted some experiments using spherical teacher model in our paper.** In Tables 1,7,8 of our paper, the rows labeled $\mathbb{S}$ indicate results obtained using spherical models either alone or as one of the teacher models. Experimental results indicate that the use of spherical teacher models provides limited improvement in the results. This situation may be improved after further refining the framework for spherical space.
>
> We will explore using teacher models from other geometric spaces to enhance the adaptability of our framework. We will also provide more detailed descriptions of the experimental setups in the paper to avoid similar misunderstandings and improve clarity.

---

> ### Author Response · Authors · 2024-08-11
>
> Dear Reviewer rr98,
>
> We thank again for your thorough review and constructive feedback. In our response to your comments, we included an analysis of our method's time and space complexity and noted that our original work have explored integrating other geometric models into the framework and presented the results. We genuinely hope you could check our responses, and kindly let us know your valuable feedback. We would be happy to provide any additional clarifications that you may need.
>
> Best regards,
>
> Authors

---

> > ### Comment · Reviewer_rr98 · 2024-08-13
> >
> > Thank the authors for their response. I will keep my score.

---

> > > ### Author Response · Authors · 2024-08-13
> > >
> > > **Dear Reviewer rr98:**
> > >
> > > Thank you for your response! We noticed that you did not mention whether our prior responses addressed your concerns. We cautiously infer that  our prior responses may not have fully alleviated your concerns regarding the performance of our framework when extended to other geometric spaces. Upon receiving your initial comments, we began conducting additional experiments to integrate spherical space into our framework. To date, we have obtained the following results (some data are from Table 1), the best results are in bold, and the suboptimal results are italic:
> > >
> > >
> > > - F1 score (%) of the **Euclidean student model** on the dataset when incorporating the spherical teacher:
> > >
> > > |                                                 | ogbn-arxiv | ogbn-proteins | Wiki-CS   | Co-Physics | Pubmed    | Citeseer  | Cora      |
> > > | ----------------------------------------------- | ---------- | ------------- | --------- | ---------- | --------- | --------- | --------- |
> > > | **Spherical Teacher**                           | 70.11      | 70.74         | 69.13     | 96.27      | 82.14     | 71.88     | 82.48     |
> > > | **Spherical + Hyperbolic Teachers**             | **71.25**  | *70.89*       | 70.07     | 96.17      | *82.23*   | 71.90     | 82.74     |
> > > | **Spherical + Euclidean Teachers**              | 70.87      | 70.56         | *70.85*   | 96.07      | 80.45     | *71.98*   | 82.89     |
> > > | **Spherical + Hyperbolic + Euclidean Teachers** | 70.75      | 70.58         | 68.70     | *96.37*    | 81.50     | 71.77     | *83.19*   |
> > > | **Hyperbolic + Euclidean Teachers**             | *70.89*    | **71.22**     | **74.17** | **96.87**  | **82.73** | **72.60** | **86.05** |
> > >
> > >
> > > - F1 score (%) of the **spherical student model** on the dataset when incorporating the spherical teacher:
> > >
> > > |                                                 | ogbn-arxiv | ogbn-proteins | Wiki-CS   | Co-Physics | Pubmed    | Citeseer  | Cora      |
> > > | ----------------------------------------------- | ---------- | ------------- | --------- | ---------- | --------- | --------- | --------- |
> > > | **Spherical Teacher**                           | 68.11      | 65.23         | 74.34     | 95.27      | 84.63     | 75.15     | 82.17     |
> > > | **Spherical + Hyperbolic Teachers**             | *70.13*    | *70.71*       | *77.98*   | 96.10      | 85.00     | *77.97*   | 84.37     |
> > > | **Spherical + Euclidean Teachers**              | 69.54      | 69.71         | 77.17     | **96.21**  | 85.08     | 77.96     | 84.13     |
> > > | **Spherical + Hyperbolic + Euclidean Teachers** | 68.53      | 70.10         | 73.10     | 96.07      | *85.72*   | 77.63     | *84.52*   |
> > > | **Hyperbolic + Euclidean Teachers**             | **70.59**  | **70.97**     | **79.24** | *96.13*    | **85.83** | **78.21** | **86.73** |
> > >
> > >
> > >
> > > The experimental results indicate that, in some cases, the distillation performance of the teacher group including spherical space teachers outperform those of the hyperbolic and Euclidean combination. This suggests that seeking additional knowledge from other geometric spaces, including spherical space, is of significant importance for improving our framework.
> > >
> > > We thank you again for your detailed review and valuable suggestions. We hope that these supplementary comments can address your concerns. We would be glad to further discuss these new experimental results with you!
> > >
> > > Best regards,
> > >
> > > Authors

---

### Official Review · Reviewer_GamE · 2024-07-13

**Soundness:** 3
**Presentation:** 2
**Contribution:** 2
**Rating:** 5
**Confidence:** 4

**Summary:**

This paper introduces a new graph distillation framework using special teacher networks to consider Euclidean and hyperbolic geometries when performing the distillation to the light-weight student GNN model.
Key techniques include Structure-Wise Knowledge Transfer (SWKT) for selecting appropriate geometric spaces and Geometric Embedding Optimization (GEO) for feature fusion across geometries.

**Strengths:**

1. It is intriguing to consider various geometrics in distillation.

2. Experimental results demonstrate effective performance on small graph node classification datasets.

3. The sensitivity analysis of hyperparameters, including thresholds like $\delta$ and $\lambda$, is insightful. It will be great if the sensitivity analysis of the weight coefficient $\beta$ could be further considered.

**Weaknesses:**

1. The paper lacks an efficiency study comparing the computational efficiency between the teacher and student models.

2. Larger dataset should be included to better evaluate the proposed distillation methods such as ogbn-arxiv and ogbn-product

3. The motivation of choosing GNN as the student network is not sufficiently justified.
While previous works focus on focus on MLP-based students, more clarity on this choice would be beneficial.

**Questions:**

1. I am curious that the input of the student model is full graphs or sampled subgraphs? How does this differ from student networks using MLPs [1].

2. Are there any baseline methods that do not incorporate various manifolds to compare with to show the effectiveness of manifold-aware approaches?


[1] Graph-less Neural Networks: Teaching Old MLPs New Tricks via Distillation.

---

> ### Author Rebuttal · Authors · 2024-08-05
>
> We sincerely thank the reviewer for the time and effort in reviewing our paper. We hope that our response can resolve your concerns. Please feel free to ask any follow-up questions.
>
> ---
> # S3:
>
> In fact, we provided sensitivity analysis of the weight coefficient $\beta$ in Figure 3 (middle).  We performed a joint analysis of the hyperparameters $\lambda$ and $\beta$ to evaluate their combined effects on our method.  Lines with different markers denote different $\beta$ amd the best value of $\beta$ is around 3.
>
> ---
> # W1:
>
> **Theoretically**：
>
> GCN's time complexity is: $O(\sum^L_{l=1}(|E|H + NH^2))$, where $N, |E|, L$ denote the number of nodes, edges, and layers, respectively, $H$ denote hidden dimension. In our paper, teacher and student models' hidden dimension is 128 and 8, respectively. Assuming graphs is sparse, which means $|E| \approx O(N)$, student is theoretically **229x** faster than their teacher models.
>
> **Empirically**:
>
> The inference times (ms) of the teacher and student models measured on our device are shown in the table below:
>
> |          | Wiki-CS | Co-Physics | Pubmed | Citeseer | Cora | Average |
> | -------- | ------- | ---------- | ------ | -------- | ---- | ------- |
> | Teacher  | 906     | 3410       | 914    | 908      | 975  | 1422    |
> | Student  | 3.9     | 12.0       | 4.4    | 4.0      | 4.4  | 4.2     |
> | Speed-up | 227x    | 284x       | 204x   | 226x     | 220x | 232x    |
>
> Our method achieves an average acceleration of **232×**.
>
> Additionally, we provided time and space complexity analysis of SWKT and GEO modules in the response to **W1 commented by reviewer rr98**. In our paper, we provide the parameter ratios (compression rate) between teacher and student for each KD method in Table 2, distillation training times and student inference time in Tables 2, 10, 11.
>
> ---
> # W2:
>
> The largest dataset used in our experiments was Coauthor Physics (495,924 edges and 34,493 nodes). We conduct further experiments on larger datasets ogbn-arxiv (1,166,243 edges, 169,343 nodes) and ogbn-proteins (39,561,252 edges, 132,534 nodes). Results show that our method consistently achieves best distillation performance on larger datasets.
>
> |               | Euclidean Teacher | Hyperbolic Teacher | FitNet | AT    | LSP   | MSKD  | VQG   | Our       |
> | ------------- | ----------------- | ------------------ | ------ | ----- | ----- | ----- | ----- | --------- |
> | Ogbn-arxiv    | 71.91             | 73.21              | 67.56  | 67.48 | 69.53 | 69.27 | 68.59 | **70.89** |
> | Ogbn-proteins | 72.83             | 69.23              | 68.71  | 68.53 | 69.45 | 70.97 | 69.54 | **71.22** |
>
> ---
> # W3:
>
> Yes, some remarkable pervious works focused on MLP-based students[1][4].
> Meanwhile, there are also significant studies in graph KD that use GNNs as the student networks, such as [2] and [3].
>
> **Our motivations for choosing GNNs as student**: We selected GNNs as the student model to explore **a new trade-off** among performance, inference speed, and compression rate. GNNs are designed for graph-based data, so GNN students can achieve performance close to their teachers even with a much small size (e.g., our method uses a GNN student model with **a hidden dimension of only 8 and ~2% of the teacher model's size**). Furthermore, GNN students demonstrate notable efficiency in inference acceleration, achieving an average **speed-up of 232x** compared to their teacher models in our method.
>
> ---
> # Q1:
>
> Our student model takes full graphs as input. The sampled subgraphs you mentioned are used by the SWKT module to enable it to extract local structure features and corresponding manifolds.
>
> **Differences between MLPs and GNNs students**:
> - GNN students are smaller, with a ~2% size of their teachers (see Table 2). MLP students and their teachers have similar sizes due to having the same number of layers and hidden dimension (see appendixes in [1][4]).
> - GNN students achieve better performance on graph data since they leverage adjacency information.
> - MLP students have faster inference speeds due to their lack of graph dependencies, with [1] showing a speed increase of 273×, our GNN students is 232× faster than their teachers.
>
> In summary, GNN students are suitable for scenarios with limited space resource, whereas MLP students are better suited for scenarios requiring higher inference speed.
>
> **Additional experiments with MLP-based student**: Our distillation method is model-agnostic. Using the same settings as [1] and [4], we test our method with GCN teachers and MLP students. Following results show that our KD method has better or similar performance with MLP students compared to GLNN [1] and VQG [4].
>
> |          | Citeseer  | Pubmed    | Cora      |
> | -------- | --------- | --------- | --------- |
> | **GLNN** | 69.23     | 74.70     | 78.97     |
> | **VQG**  | 74.59     | 77.13     | **81.15** |
> | **Our**  | **76.00** | **77.49** | 81.08     |
>
> (part of results from [4])
>
> ---
> # Q2:
>
> In fact, except our method, **all baseline methods do not incorporate various manifolds, as we are the first to introduce a KD framework using multiple geometric teachers**. In Table 1, rows marked as $\mathbb{E}$ show results for baseline methods in their original versions without additional manifolds information, while rows marked as $\mathbb{B}$ and $\mathbb{S}$ show results after adapting to hyperbolic and spherical spaces, respectively. This adaptation ensures a fair comparison and evaluates the SWKT and GEO modules. We apologize for any confusion caused by unclear descriptions about Table 1. We will improve the presentation of the experimental section to enhance clarity.
>
> ---
> References:
>
> [1] Graph-less Neural Networks: Teaching Old MLPs New Tricks via Distillation, ICLR’22
>
> [2] Distilling Knowledge from Graph Convolutional Networks, CVPR’20
>
> [3] Boosting Graph Neural Networks via Adaptive Knowledge Distillation, AAAI’23
>
> [4] VQGraph: Rethinking Graph Representation Space for Bridging GNNs and MLPs, ICLR’24

---

> ### Author Response · Authors · 2024-08-11
>
> Dear Reviewer GamE,
>
> We appreciate your thorough review and constructive feedback. We have provided an analysis of the computational efficiency of our method and further validated its performance on the ogbn-arxiv and ogbn-proteins datasets. We have analyzed the differences between GNN and MLP student models and clarified potential misunderstandings about the baseline methods. We would be grateful for the opportunity to discuss whether your concerns have been fully addressed. Please let us know if you still have any unclear part of our work.
>
> Best regards,
>
> Authors

---

> > ### Comment · Reviewer_GamE · 2024-08-13
> > **Official comments by Reviewer GamE**
> >
> > Thanks for the authors' effort to address my concerns. I have raised my score.

---

> > > ### Author Response · Authors · 2024-08-13
> > > **Thank you!**
> > >
> > > We are glad that our initial rebuttal addressed your concerns well!
> > >
> > > Graph data may exhibit geometric heterogeneity (e.g., grid or tree structures) across different local structures. Therefore, we are committed to matching appropriate embedding geometric spaces through meticulous analysis and extraction of local structural features to enhance graph representation quality. We validated the effectiveness of this paradigm through knowledge distillation tasks.
> > >
> > > We are delighted that you find our approach of consider various geometries in distillation intriguing, and that the experimental results demonstrate its effectiveness. We fully agree that conducting experiments on larger datasets and analyzing the selection motivations and computational efficiency of the student models will enhance the quality of our paper. In our initial rebuttal, we conducted experiments on larger datasets and provided a comparative analysis of the computational efficiency between the teacher and student models, as well as explained the rationale behind the selection of student model architectures. We will incorporate these additions into the revised version of our paper as your suggestions! Thank you again for your insightful comments and valuable suggestions!
> > >
> > > Best regards,
> > >
> > > Authors

---

### Official Review · Reviewer_Z3B3 · 2024-07-13

**Soundness:** 3
**Presentation:** 4
**Contribution:** 4
**Rating:** 6
**Confidence:** 3

**Summary:**

This paper proposes a novel methodology to distill graph neural network (GNN) integrating the various geometries (e.g. Euclidean and hyperbolic). The author first utilizes a structure-wise knowledge transfer module with multiple teacher models from distinct geometric properties.  Then, the authors demonstrate a geometric embedding optimization to guide a student model optimizing cross-geometric space. The authors show the efficacy of their approach through experiments on node classification and link prediction with various datasets compared with existing state-of-the-art methods.

**Strengths:**

1. The paper is well written.
2. The authors have extensive experiments including the comparison to the state-of-the-art and ablation studies on their new contributing components.

**Weaknesses:**

1. Some questions exist about the authors' empirical results. Please see the questions section.

**Questions:**

1. Are Table 1 results all reproduced by the authors? If so, the reported performance of other baselines seems significantly lower than the numbers reported in other papers. For example, MSKD (Zhang et al.) reports an 89.67 F1-Score with an 8.2 ratio on the CiteSeer node classification task. On the other hand, the MSKD NC F1-Score in Table 1 is only 71.42.
2. Following up with 1, GAT results in Table 7 on CiteSeer are significantly lower than the MSKD paper reported. For example, the teacher model trained on CiteSeer with Euclidean geometries only achieves an F1-score of 75.24, while the MSKD paper reports their student model achieves a 95.47 F1-Score with a 17.6 ratio!
3. Can you do the hyperparameter tuning on the training such that the teacher's performance on Citeseer is close to what MSKD (Zhang et al.) has reported? In distillation literature, experimenting with the very well-trained teacher model is important. If not, the gain from the student model could have been illusory since the performance gap could have been larger if the teacher had trained with better hyperparameters.
4. VQG authors (Yang et al.) claim that VQGraph has a significant advantage in the inference speed (828x compared to GNN), but the Table 2 inference speeds report all the same inference speeds on FitNet, AT, LSP, MSK, VQG, and the author's method on NC. I am skeptical whether these are the right numbers.
5. Also, VQG may not be the right candidate to compare since VQG tackles a different problem, a GNN to MLP distillation.

**Limitations:**

Overall, the authors' experiments show their new contributions can improve the GNN models in an agnostic way. However, some numerical performance results from the authors are too low (including both teacher and student) compared to the other papers reported. After the authors match their teacher's performance to the other papers (e.g. CiteSeer from MSKD by Zhang et al.) and observe good student performance with a small gap to the teacher + state-of-the-art results, I am happy to raise the score.

---

> ### Author Rebuttal · Authors · 2024-08-05
>
> We sincerely thank the reviewer for the time and effort in reviewing our paper. We take all comments seriously and try our best to address every raised concerns. Please feel free to ask any follow-up questions.
>
> ---
> # Q1:
>
> Yes, our experimental settings are strictly following the original papers. We used same pre-trained teachers, which were the best-performing models obtained by extensive hyperparameter tuning.
>
> MSKD[1] is an amazing baseline method that introduces multi-scale teachers, providing more comprehensive information to enhance distillation efficiency. Regarding the impressive performance of the teacher model in MSKD  (GCN 89.67%, GAT 95.47%) on Citeseer, despite extensive efforts, we were unable to make our GCN teacher has the similar performance. Since the authors did not release Citeseer dataset used, the significant performance differences may be due to variations in dataset preprocessing or dataset versions.
>
> We believe that we have obtained a well-trained and effective GCN teacher model on Citeseer, since our GCN teacher have a comparable performance (73.97%) with some latest works which use Citeseer. VQG (one of our baseline method, ICLR'24) [2] shows that the GCN teacher model of performs at 70.49% on Citeseer. Additionally, reported F1 scores for node classification with GCN on Citeseer are 71.0% in [3] (NIPS’23) , 72.9% in [4] (NIPS’23), and 73.18% in [5] (WWW’23).
>
> ---
> # Q2:
>
> We also believe that we have obtained a well-trained and effective GAT teacher on Citeseer. Reported F1 scores with GAT on Citeseer are 73.00% in [4] (NIPS’23) ,76.63% in [5] (WWW’23) . These results are comparable to our GAT teacher's performance (75.24%)
>
> ---
> # Q3：
>
> Unfortunately, due to a lack of detailed experimental information and specific dataset versions, we could not replicate MSKD teachers' performance on Citeseer.
>
> We agree that using a well-trained teacher model is very important in KD. Therefore, we conducted extensive hyperparameter tuning to ensure our teacher models achieved near-optimal performance on each dataset, comparable to the best results reported in latest other graph-related studies [2][3][4][5].
>
> Moreover, all methods employed the same student architecture. Thus, the concern that insufficient training of the teacher model may result in illusory performance of the student model should not appear in our experiments.
>
> ---
> # Q4:
>
> All runtime data were recorded using Python's datetime library.
> In our experiments, we replaced the student model of VQG with GCN to maintain consistency with other baseline methods  This adjustment is likely the primary reason why all students, including in VQG, exhibit similar inference times.
>
> We analyzed the speed-up of the student compared to their teachers in our method:
>
> **Theoretically**：
>
> GCN layer's time complexity is: $O(\sum^L_{l=1}(|E|H + NH^2))$
>
> where $N, |E|, L$ denote the number of nodes, edges, and layers, respectively, $H$ denote the hidden layer dimension. The teacher and student models have hidden dimension of 128 and 8, respectively.
>
> Assuming the graph is sparse, which means $|E| \approx O(N)$. the student model is theoretically **229x** faster than the teacher model about.
>
> **Empirically**:
>
> The inference times (ms) of the teacher and student models measured by  datetime library are shown below:
>
> |          | Wiki-CS | Co-Physics | Pubmed | Citeseer | Cora | Average |
> | -------- | ------- | ---------- | ------ | -------- | ---- | ------- |
> | Teacher  | 906     | 3410       | 914    | 908      | 975  | 1422    |
> | Student  | 3.98    | 12.0       | 4.46   | 4.01     | 4.43 | 4.22    |
> | Speed-up | 227x    | 284x       | 204x   | 226x     | 220x | 232x    |
>
> Our method achieves a notable acceleration with **an average speedup of 232×**. While the different student architectures result in lower speedup compared to VQG, our method's student  achieves a high compression rate (~2%, see Table 2), whereas VQG's MLP students and teachers are similar in size (see appendix in [2]).
> We chose GNN as the student model due to considerations of the trade-off between performance and inference speed. Please refer our responses to **Reviewer GamE's W3 and Q1** for more details.
>
> ---
> # Q5:
>
> Yes, VQG[6] was originally proposed to address the GNN-to-MLP conversion. In our experiments, we replaced the student model of VQG with GCN to maintain consistency with other baseline method.
>
> We use GCN as the teacher model and replaced the student model with MLP same as VQG's student for a comparison with VQG and another MLP-based graph KD method GLNN[7]. We keep same settings with [6], and the results shown in the following table demonstrate that our method achieves effective distillation even with MLP-based student networks.
>
> |          | Citeseer  | Pubmed    | Cora      |
> | -------- | --------- | --------- | --------- |
> | **GLNN** | 69.23     | 74.70     | 78.97     |
> | **VQG**  | 74.59     | 77.13     | **81.15** |
> | **Our**  | **76.00** | **77.49** | 81.08     |
>
> (part of results from [6])
>
> ---
> In the responses, we explained the reasons why our teacher models are being well-trained. Besides, all methods used same settings and pre-trained teacher models, with results averaged over 10 trials. Based on these solid results, our method achieved sota performance, we hope this can change your opinion.
>
> ---
> References:
>
> [1] Multi-Scale Distillation from Multiple Graph Neural Networks , AAAI’22
>
> [2] VQGraph: Rethinking Graph Representation Space for Bridging GNNs and MLPs, ICLR’24
>
> [3] On Class Distributions Induced by Nearest Neighbor Graphs for Node Classification of Tabular Data, NIPS’23
>
> [4] Re-Think and Re-Design Graph Neural Networks in Spaces of Continuous Graph Diffusion Functionals, NIPS’23
>
> [5] GIF: A General Graph Unlearning Strategy via Influence Function, WWW’23
>
> [6] VQGraph: Rethinking Graph Representation Space for Bridging GNNs and MLPs, ICLR’24
>
> [7] Graph-less Neural Networks: Teaching Old MLPs New Tricks via Distillation, ICLR’22

---

> ### Author Response · Authors · 2024-08-11
>
> Dear Reviewer Z3B3,
>
> We sincerely thank you for your time and effort in reviewing our paper. In responses to your comments, we reviewed relevant works and demonstrated that our pre-trained teacher models are sufficiently trained on the Citeseer dataset. We analyzed the acceleration ratio of the student model compared to the teacher model from both theoretical and practical perspectives. Additionally, we replaced our student model with an MLP model of the same architecture as used in the VQG method and conducted comparative experiments. We would be grateful for the opportunity to discuss whether your concerns have been fully addressed. Please let us know if you still have any unclear part of our work.
>
> Best regards,
>
> Authors

---

> > ### Comment · Reviewer_Z3B3 · 2024-08-13
> >
> > Thank you for your response. After thoroughly reviewing it and considering the results of the additional experiments, most of my concerns have been addressed. I raised my score.

---

> ### Author Response · Authors · 2024-08-13
> **Thank you!**
>
> Thank you for your positive feedback and we are glad to know that our rebuttal and new experiments have addressed most of your concerns!
>
> Graph data may exhibit geometric heterogeneity (e.g., grid or tree structures) across different local structures. Therefore, we are committed to matching appropriate embedding geometric spaces through meticulous analysis and extraction of local structural features to enhance graph representation quality. We validated the effectiveness of this paradigm through knowledge distillation tasks.
>
> We are pleased to learn that you found our paper is well-written and that the experimental results validate the effectiveness of our agnostic cross-geometric framework and modules. In our initial rebuttal, we analyzed the speedup achieved by our method from both theoretical and practical perspectives and compared our method with GNN-to-MLP methods by replacing the student model of our method with MLP. We will ensure that these important details are included in the revised version. Thank you again for your insightful comments and valuable suggestions!
>
> Best regards,
>
> Authors

---

### Official Review · Reviewer_ogwd · 2024-07-13

**Soundness:** 3
**Presentation:** 3
**Contribution:** 2
**Rating:** 5
**Confidence:** 2

**Summary:**

This paper presents a cross-geometric graph knowledge distillation method for graph neural networks. This method employs multiple teacher models, each generating different embeddings with distinct geometric properties, such as Euclidean, hyperbolic, and spherical spaces. The student model is based on Euclidean space. Two modules, Structure-Wise Knowledge Transfer (SWKT) and Geometric Embedding Optimization (GEO), are proposed to enhance performance. To evaluate the proposed approach, distillation experiments are conducted on node classification (NC) and link prediction (LP) tasks across various types of graph data.

**Strengths:**

1. This work presents a novel cross-geometric knowledge distillation framework. Distilling knowledge from Euclidean and hyperbolic geometries in a space-mixing fashion is a new approach for me.
2. A fine-grained analysis of the subgraphs is provided.
3. Experiments show better results compared to previous knowledge distillation methods.

**Weaknesses:**

1. In the presented cross-geometric knowledge distillation framework, the student model learns from multiple teacher models with different geometric information. Properly training all these teacher models is challenging. It is unclear how to ensure the student model's reliability if some teacher models fail.

2. The student model operates only in Euclidean space, raising questions about whether other student models, such as those in hyperbolic space, can achieve similar results.

3. All experiments are conducted solely on GCN, with no results provided for other graph architectures.

4. It is unclear if a student model using the same architecture as the teacher models, within the cross-geometric knowledge distillation framework, can achieve better performance than the teacher models.

**Questions:**

1. If one or more teacher models fail, how does this impact the performance of the student model?
2. Can the student model operate in other geometric spaces, such as hyperbolic space?
3. Can the proposed framework work with other architectures, such as Graph Transformer Networks [1]?
4. If the student model uses the same architecture as the teacher models, how does it perform?

[1] Seongjun Yun, Minbyul Jeong, Raehyun Kim, Jaewoo Kang, Hyunwoo J. Kim:
Graph Transformer Networks. NeurIPS 2019

**Limitations:**

This work has discussed the potential limitations in their paper:

---

> ### Author Rebuttal · Authors · 2024-08-06
>
> We sincerely thank the reviewer for the time and effort in reviewing our paper. We hope that our response can resolve your concerns. Please feel free to ask any follow-up questions.
>
> ---
> # W1 & Q1:
>
> It is true that the failure of one or more teacher models could potentially impact the student model’s performance, we have implemented several mechanisms in our method to mitigate this risk:
> - **Ensemble Learning:** Using multiple teacher models that capture different geometric properties provides redundancy and robustness. If one model fails, the others still contribute valuable insights, minimizing the impact on the student model.
> - **Geometric Optimization Network:** GEO dynamically adjusts the weight of information from each teacher model based on the loss function, reducing the influence of any underperforming model and ensuring the student model receives the most reliable information.
>
> **Additional experiments**：
> We designed various experimental strategies to assess the impact of failing teachers on students:
>
> S1: Train student models without KD.
> S2: Train student models with the best-tuned teacher model.
> S3: Train student models with an underperforming teacher model.
> S4: Train student models with an untrained teacher model.
> S5: Train student models with all untrained teacher models.
> Note: All methods except MSKD and ours use a single teacher model,so their S5 f1 scores are N/A.
>
> Teachers' F1 score (%) of different experimental strategies:
>
> |                    | S2    | S3    | S4    | S5    |
> | ------------------ | ----- | ----- | ----- | ----- |
> | Euclidean Teacher  | 86.98 | 66.09 | 36.61 | 25.55 |
> | Hyperbolic Teacher | 90.90 | 64.86 | 52.58 | 30.12 |
>
> NC f1 score (%) of student models on Cora under different experimental strategies:
>
> | Methods | S1↑   | S2↑   | S3↑   | S4↑   | S5↑   | Std(S2-S4) |
> | ------- | ----- | ----- | ----- | ----- | ----- | ---------- |
> | FitNet  | 82.06 | 80.32 | 56.27 | 53.81 | N/A   | 11.95      |
> | AT      | 82.06 | 80.49 | 63.39 | 51.11 | N/A   | 12.04      |
> | LSP     | 82.06 | 83.34 | 71.33 | 56.76 | N/A   | 10.86      |
> | MSKD    | 82.06 | 82.48 | 81.82 | 78.62 | 21.62 | 1.68       |
> | VQG     | 82.06 | 83.02 | 70.02 | 56.02 | N/A   | 11.02      |
> | Our     | 82.06 | 86.05 | 85.26 | 81.33 | 22.85 | 2.06       |
>
> Except S5, which teachers have an average performance of only 30%, our method’s distilled student models consistently maintain stable performance even when some teacher models fail.
>
> ---
> # W2 & Q2:
>
> Yes, the student model can operate in other geometric spaces. At first, we chose a Euclidean student model to combine hyperbolic accuracy benefits with Euclidean efficiency and stability. Our framework is model-agnostic, allowing replacement of the student model with other neural networks.
>
> To validate student on various geometric spaces, we tested NC F1 scores (%) on the Cora dataset. Euclidean and hyperbolic teachers' F1 score is 86.98% and 90.90%.
>
> |                    | FitNet | AT    | LSP   | MSKD  | VQG   | Our       |
> | ------------------ | ------ | ----- | ----- | ----- | ----- | --------- |
> | Euclidean Student  | 80.32  | 80.49 | 83.34 | 82.48 | 83.02 | 86.05     |
> | Hyperbolic Student | 86.73  | 86.00 | 87.96 | 88.21 | 88.45 | 90.42     |
> | Spherical Student  | 75.92  | 83.54 | 84.77 | 85.26 | 83.54 | 86.73     |
> | Average            | 80.99  | 83.34 | 85.35 | 85.31 | 85.00 | **87.73** |
>
> ---
> # W3 & Q3:
>
> Our framework is model-agnostic and works with various model architectures.
> In fact, we have conducted experiments with alternative architectures. We apologize if our results for replacing GCN with GAT, originally presented in Table 7 in the appendix due to space constraints, escaped the reviewer's attention.
>
> We also followed your suggestion and replaced the Euclidean teacher model with the Graph Transformer Network (GTN)[1].
>
> |        | GTN Teacher | Hyperbolic Teacher | GTN Student | GCN Student | GTN Student Inference time | GCN Student Inference time |
> | ------ | ----------- | ------------------ | ----------- | ----------- | -------------------------- | -------------------------- |
> | WikiCS | 82.74       | 81.83              | 82.48       | 74.26       | 15.34ms                    | 3.98ms                     |
> | Cora   | 87.87       | 90.90              | 86.37       | 86.24       | 17.67ms                    | 4.43ms                     |
>
> GTN teacher has 4 layers and a hidden dimension of 128. Specifically, during distillation, student model's $l$ layers match the last $l$ layers of teacher accordingly. The GTN and HGCN teacher output intermediate representations from each layer to the SWKT module for local subgraph structure extracting and selection. These distributions are then optimized by GEO module. Then, these features extracted from the optimized cross-geometric intermediate representations are transferred to students via the corresponding loss function. Additionally, traditional KD loss is computed from the logits output by both the teacher and student models.
>
> ---
> # W4 & Q4:
>
> As your suggestion, we conducted experiments using student models with the same architecture as the teacher models in our method. Following results are NC F1 scores (%) on the Cora dataset. Euclidean and hyperbolic teachers' F1 score is 86.98% and 90.90%.
>
> |                    | FitNet | AT    | LSP   | MSKD  | VQG   | Our   |
> | ------------------ | ------ | ----- | ----- | ----- | ----- | ----- |
> | Student in paper   | 80.32  | 80.49 | 83.34 | 82.48 | 83.02 | 86.05 |
> | Euclidean student  | 86.73  | 86.24 | 86.98 | 86.98 | 86.49 | 87.47 |
> | hyperbolic student | N/A    | N/A   | N/A   | N/A   | N/A   | 90.42 |
>
> Compared to the results presented in Table 1 of our paper, student models now even outperform some teacher models, but using the same architecture as the teacher makes student models larger and slower, limiting their suitability for resource-constrained devices.
>
> ---
> References:
>
> [1] Graph Transformer Networks. NeurIPS'19

---

> ### Author Response · Authors · 2024-08-11
>
> Dear Reviewer ogwd,
>
> We sincerely thank you for your time and effort in reviewing our paper. In responses to your comments, we have carefully designed and conducted experiments and analyses, which we believe have covered your concerns. We would be grateful for the opportunity to discuss whether your concerns have been fully addressed. Please let us know if you still have any unclear part of our work.
>
> Best regards,
>
> Authors

---

> > ### Comment · Reviewer_ogwd · 2024-08-13
> >
> > Thank you for the response. I have carefully reviewed it and the results of the new experiments have addressed most of my concerns. I am pleased to increase my score.

---

> > > ### Author Response · Authors · 2024-08-13
> > > **Thank you!**
> > >
> > > We sincerely appreciate your valuable reviews and are glad to know that our rebuttal and new experiments have addressed most of your concerns.
> > >
> > > Graph data may exhibit geometric heterogeneity (e.g., grid or tree structures) across different local structures. Therefore, we are committed to matching appropriate embedding geometric spaces through meticulous analysis and extraction of local structural features to enhance graph representation quality. We validated the effectiveness of this paradigm through knowledge distillation tasks.
> > >
> > > We are delighted to see that you consider our method, which combines the advantages of teacher models from different geometric spaces and analyzes local subgraph features from a fine-grained perspective for knowledge distillation, to be an innovative distillation paradigm. We fully agree that further validating the robustness of our framework (e.g., cases where some teacher models fail) and its model-agnostic nature (e.g., changing the architecture of teacher and student models) will strengthen our work. We conducted some experiments in our initial response to provide support and will incorporate these results into the experimental section in the revised version as your suggestions! Thank you again for your insightful comments and valuable suggestions!
> > >
> > > Best regards,
> > >
> > > Authors

---

### Author Rebuttal · Authors · 2024-08-07

Dear Reviewers,

We sincerely thank the reviewers for the time and effort in reviewing our paper. We take all comments seriously and try our best to address every raised concerns. Please feel free to ask any follow-up questions.

It is encouraging to learn that the reviewers stated that our method, which leverages the advantages of different geometric spaces for graph knowledge distillation, is novel (ogwd, rr98) and intriguing (GamE). Reviewers also acknowledged that extensive (Z3B3) experiments conducted on various datasets (ogwd, Z3B3) show that our method outperforms baseline methods (ogwd, Z3B3, rr98). Reviewer Z3B3 kindly noted that the paper is well written.

Since the reviewers focused on empirical performance of our method, we would like to highlight our **theoretical contributions** here. To address the challenge that real-world graphs often exhibit geometrically heterogeneous characteristics, we have introduced **a novel model-agnostic cross-geometric knowledge distillation framework**. This framework, for the first time, leverages the advantages of various geometric spaces to offer high-quality guidance to the student model. Our dual-teacher setup and adaptive GEO module ensure that our method remains effective even under adverse conditions (such as partial teacher fail). Furthermore, we introduce **a new fine-grained distillation perspective** that evaluates embeddings, extracts knowledge, and transfers it as hints to the student model based on local subgraphs.

In the past week, we carefully improved the experiments (utilizing all available computational resources), clarified various aspects, and expanded our discussions to address the concerns, questions, and requests from all four reviewers. **In summary, we have implemented the following improvements**:

- We analyzed the computational efficiency of student models compared to teacher models in our method, examined the inference acceleration achieved by student models, and validated these findings through experiments. (in response to reviewer Z3B3’s Q4, Game’s W1)

- We provided a complete analysis of the time and space complexity of our method. (in response to reviewer rr98's W1)

- We conducted distillation experiments with non-Euclidean student models. (in response to reviewer ogwd’s Q2)

- We conducted distillation experiments by replacing the teacher model with graph transformer. (in response to reviewer ogwd’s Q3)

- We replaced the student model with MLPs, conducted distillation experiments, and compared the results with method of using MLPs as student. (in response to reviewer Z3B3’s Q5, Game’s Q1)

- We conducted distillation experiments on larger datasets, ogbn-arxiv and ogbn-proteins. (in response to reviewer Game’s W2)

- We analyzed the impact of fail teacher model on distillation results and designed experiments to validate this. (in response to reviewer ogwd’s Q1)

- We clarified the motivation for choosing GNN as the student model and its differences from MLP students. (in response to reviewer Game’s W3 and Q1)

- We conducted distillation experiments where the teacher and student models had identical architecture. (in response to reviewer ogwd’s Q4)

---

> ### Author Response · Authors · 2024-08-14
> **General Response**
>
> We sincerely thank the reviewers for their time and effort in reviewing our paper and discussing with us. We appreciate the AC, SAC, and PC for their diligent efforts in organizing and overseeing the paper review process. We are delighted to see that during the discussion phase, Reviewer ogwd, Z3B3, GamE stated that our responses and new experiments have successfully addressed most of their concerns, and that all reviewers ultimately have a positive attitude toward our paper. We will ensure that these analyses and experimental results are included in the revised version of the paper.
>
> Best regards,
>
> Authors

---

### Decision · Program_Chairs · 2024-09-25

**Decision:**

Accept (poster)

**Comment:**

This paper proposes a graph knowledge distillation framework for transferring knowledge between different graph geometries. The framework distills knowledge from multiple teacher GNNs, each aware of a specific geometry, into a smaller student GNN. This approach not only improves the quality of graph representations but also accelerates inference speed. The authors successfully addressed the reviewers' concerns during the author-reviewer discussion phase, and at the end, all reviewers unanimously support acceptance. AC also finds that this paper has several strengths, including its novelty, extensive experimental results, and superior performance.

Therefore, AC recommends acceptance and highly encourages the authors to incorporate all experimental results and discussions from the discussion phase into the final manuscript, as this would strengthen the contributions of this paper.